# Deep transcriptome annotation enables the discovery and functional characterization of cryptic small proteins

Sondos Samandi[1,2†], Annie V Roy[1,2†], Vivian Delcourt[1,2,3], Jean-François Lucier[4,5], Jules Gagnon[4,5], Maxime C Beaudoin[1,2], Benoît Vanderperre[1], Marc-André Breton[1], Julie Motard[1,2], Jean-François Jacques[1,2], Mylène Brunelle[1,2], Isabelle Gagnon-Arsenault[2,6,7], Isabelle Fournier[3], Aida Ouangraoua[8], Darel J Hunting[9], Alan A Cohen[10], Christian R Landry[2,6,7], Michelle S Scott[1], Xavier Roucou[1,2]*

[1]Department of Biochemistry, Université de Sherbrooke, Sherbrooke, Canada; [2]PROTEO, Québec Network for Research on Protein Function, Structure and Engineering, Québec, Canada; [3]INSERM U1192, Laboratoire Protéomique, Réponse Inflammatoire & Spectrométrie de Masse (PRISM) F-59000 Lille, Université de Lille, Lille, France; [4]Department of Biology, Université de Sherbrooke, Québec, Canada; [5]Center for Scientific computing, Information Technologies Services, , Université de Sherbrooke, Québec, Canada; [6]Département de biochimie, microbiologie et bioinformatique, Université Laval, Québec, Canada; [7]IBIS, Université Laval, Québec, Canada; [8]Department of Computer Science, Université de Sherbrooke, Québec, Canada; [9]Department of Nuclear Medicine and Radiobiology, Université de Sherbrooke, Québec, Canada; [10]Department of Family Medicine, Université de Sherbrooke, Québec, Canada

*For correspondence:
xavier.roucou@usherbrooke.ca

[†]These authors contributed equally to this work

Competing interests: The authors declare that no competing interests exist.

**Abstract** Recent functional, proteomic and ribosome profiling studies in eukaryotes have concurrently demonstrated the translation of alternative open-reading frames (altORFs) in addition to annotated protein coding sequences (CDSs). We show that a large number of small proteins could in fact be coded by these altORFs. The putative alternative proteins translated from altORFs have orthologs in many species and contain functional domains. Evolutionary analyses indicate that altORFs often show more extreme conservation patterns than their CDSs. Thousands of alternative proteins are detected in proteomic datasets by reanalysis using a database containing predicted alternative proteins. This is illustrated with specific examples, including altMiD51, a 70 amino acid mitochondrial fission-promoting protein encoded in *MiD51/Mief1/SMCR7L*, a gene encoding an annotated protein promoting mitochondrial fission. Our results suggest that many genes are multicoding genes and code for a large protein and one or several small proteins.
DOI: https://doi.org/10.7554/eLife.27860.001

## Introduction

Current protein databases are cornerstones of modern biology but are based on a number of assumptions. In particular, a mature mRNA is predicted to contain a single CDS; yet, ribosomes can select more than one translation initiation site (TIS) (*Ingolia et al., 2011*; *Lee et al., 2012*; *Mouilleron et al., 2016*) on any single mRNA. Also, minimum size limits are imposed on the length of CDSs, resulting in many RNAs being mistakenly classified as non-coding (ncRNAs) (*Pauli et al., 2014*; *Anderson et al., 2015*; *Zanet et al., 2015*; *Nelson et al., 2016*; *Bazzini et al., 2014*; *Ji et al.,*

**eLife digest** Proteins are often referred to as the workhorses of the cell, and these molecules affect all aspects of human health and disease. Thus, deciphering the entire set of proteins made by an organism is often an important challenge for biologists.

Genes contain the instructions to make a protein, but first they must be copied into a molecule called an mRNA. The part of the mRNA that actually codes for the protein is referred to as an open reading frame (or ORF for short). For many years, most scientists assumed that, except for in bacteria, each mature mRNA in an organism has just a single functional ORF, and that this was generally the longest possible ORF within the mRNA. Many also assumed that RNAs copied from genes that had been labelled as "non-coding" or as "pseudogenes" did not contain functional ORFs.

Yet, new ORFs encoding small proteins were recently discovered in RNAs (or parts of RNA) that had previously been annotated as non-coding. Working out what these small proteins actually do will require scientists being able to find more of these overlooked ORFs.

The RNAs produced by many organisms – from humans and mice to fruit flies and yeast – have been catalogued and the data stored in publicly accessible databases. Samandi, Roy et al. have now taken a fresh look at the data for nine different organisms, and identified several thousand examples of possibly overlooked ORFs, which they refer to as "alternative ORFs". This included more than 180,000 from humans.

Further analysis of other datasets that captured details of the proteins actually produced in human cells uncovered thousands of small proteins encoded by the predicted alternative ORFs. Many of the so-called alternative proteins also resembled parts of other proteins that have a known activity or function. Lastly, Samandi, Roy et al. focused on two alternative proteins and showed that they both might affect the activity of the proteins coded within the main ORF in their respective genes.

These findings reveal new details about the different proteins encoded within the genes of humans and other organisms, including that many mRNAs encode more that one protein. The implications and applications of this research could be far-reaching, and may help scientists to better understand how genes work in both health and disease.

DOI: https://doi.org/10.7554/eLife.27860.002

---

*2015*; *Prabakaran et al., 2014*; *Slavoff et al., 2013*). As a result of these assumptions, the size and complexity of most eukaryotic proteomes have probably been greatly underestimated (*Andrews and Rothnagel, 2014*; *Landry et al., 2015*; *Fields et al., 2015*; *Saghatelian and Couso, 2015*). In particular, few small proteins (defined as of 100 amino acids or less) are annotated in current databases. The absence of annotation of small proteins is a major bottleneck in the study of their function and their roles in health and disease. This is further supported by classical and recent examples of small proteins of functional importance, for instance many critical regulatory molecules such as the F0 subunit of the F0F1-ATPsynthase (*Stock et al., 1999*), the sarcoplasmic reticulum calcium ATPase regulator phospholamban (*Schmitt et al., 2003*), and the key regulator of iron homeostasis hepcidin (*Nemeth et al., 2004*). This limitation also impedes our understanding of the process of origin of new genes, which are thought to contribute to evolutionary innovations. Because these genes generally code for small proteins (*Carvunis et al., 2012*; *Schlötterer, 2015*; *McLysaght and Hurst, 2016*; *Sabath et al., 2012*), they are difficult to unambiguously detect by proteomics and in fact are impossible to detect if they are not included in proteomics databases.

Functional annotation of ORFs encoding small proteins is particularly challenging since an unknown fraction of small ORFs may occur by chance in the transcriptome, generating a significant level of noise (*Landry et al., 2015*). However, given that many small proteins have important functions and are ultimately one of the most important sources of functional novelty, it is time to address the challenge of their functional annotations (*Landry et al., 2015*).

We systematically reanalyzed several eukaryotic transcriptomes to annotate previously unannotated ORFs which we term alternative ORFs (altORFs), and we annotated the corresponding hidden proteome. Here, altORFs are defined as potential protein-coding ORFs in ncRNAs, in UTRs or in

different reading frames from annotated CDSs in mRNAs (*Figure 1a*). For clarity, predicted proteins translated from altORFs are termed alternative proteins and proteins translated from annotated CDSs are termed reference proteins.

Our goal was to provide functional annotations of alternative proteins by (1) analyzing relative patterns of evolutionary conservation between alternative and reference proteins and their

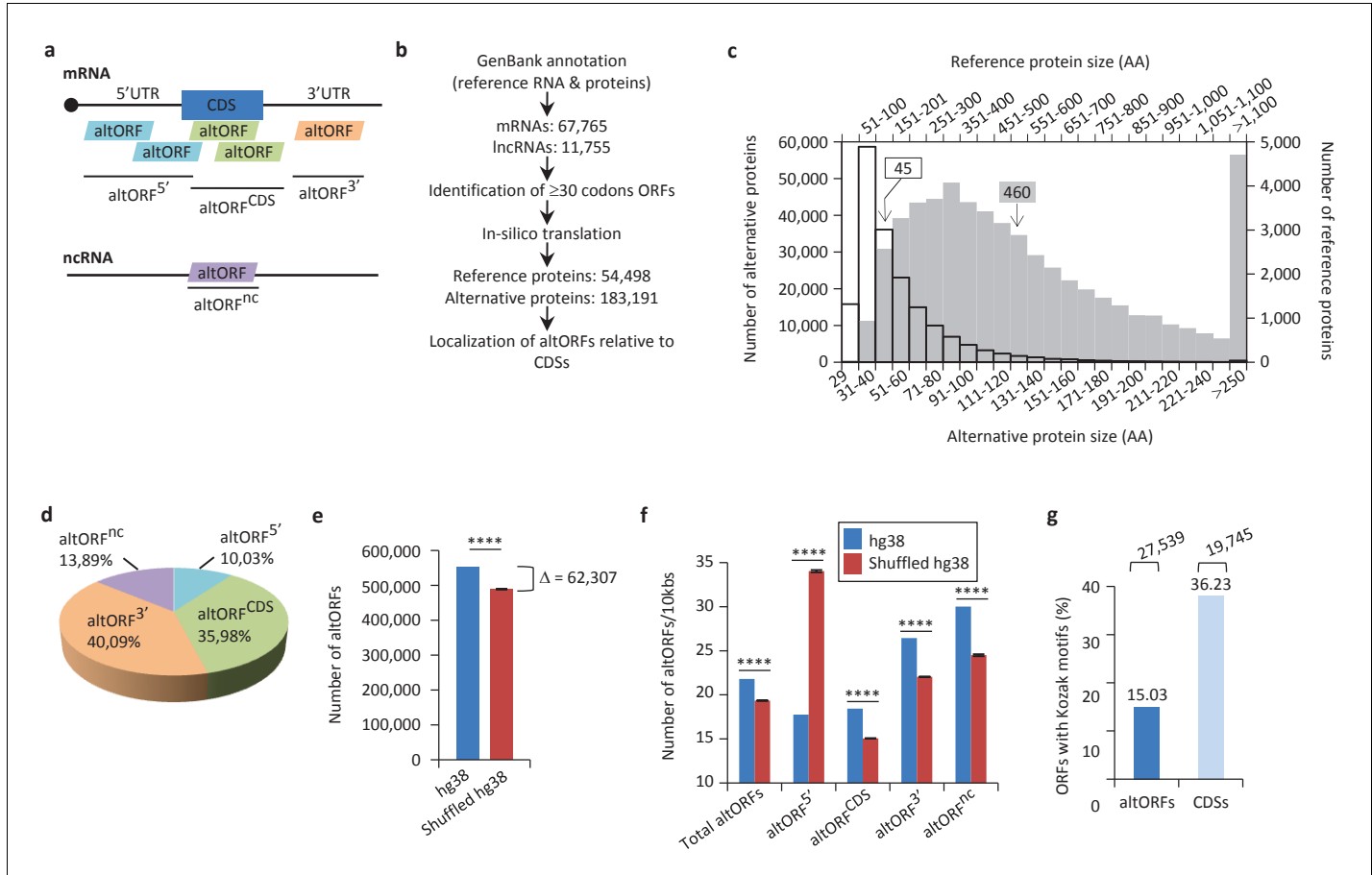

**Figure 1.** Annotation of human altORFs. (a) AltORF nomenclature. AltORFs partially overlapping the CDS must be in a different reading frame. (b) Pipeline for the identification of altORFs. (c) Size distribution of alternative (empty bars, vertical and horizontal axes) and reference (grey bars, secondary horizontal and vertical axes) proteins. Arrows indicate the median size. The median alternative protein length is 45 amino acids (AA) compared to 460 for the reference proteins. (d) Distribution of altORFs in the human hg38 transcriptome. (e, f) Number of total altORFs (e) or number of altORFs/10kbs (f) in hg38 compared to shuffled hg38. Means and standard deviations for 100 replicates obtained by sequence shuffling are shown. Statistical significance was determined by using one sample t-test with two-tailed p-values. ****$p<0.0001$. (g) Percentage of altORFs with an optimal Kozak motif. The total number of altORFs with an optimal Kozak motif is also indicated at the top.

DOI: https://doi.org/10.7554/eLife.27860.003

The following source data and figure supplements are available for figure 1:

**Source data 1.** Annotation of human altORFs.
DOI: https://doi.org/10.7554/eLife.27860.008

**Figure supplement 1.** 10% of altORFs are present in different classes of repeats.
DOI: https://doi.org/10.7554/eLife.27860.004

**Figure supplement 1—source data 1.** 10% altORFs are present in different classes of repeats.
DOI: https://doi.org/10.7554/eLife.27860.005

**Figure supplement 2.** The proportion of altORFs with a translation initiation site (TIS) with a Kozak motif in hg38 is significantly different from 100 shuffled hg38 transcriptomes.
DOI: https://doi.org/10.7554/eLife.27860.006

**Figure supplement 2—source data 1.** Proportion of altORFs with a Kozak motif in hg38 and shuffled hg38.
DOI: https://doi.org/10.7554/eLife.27860.007

corresponding coding sequences; (2) estimating the prevalence of alternative proteins both by bioinformatics analysis and by detection in large experimental datasets; (3) detecting functional signatures in alternative proteins; and (4) testing the function of some alternative proteins.

## Results

### Prediction of altORFs and alternative proteins

We predicted a total of 539,134 altORFs compared to 68,264 annotated CDSs in the human transcriptome (*Figure 1b*, *Table 1*). Because identical ORFs can be present in different RNA isoforms transcribed from the same genomic locus, the number of unique altORFs and CDSs becomes 183,191 and 54,498, respectively. AltORFs were also predicted in other organisms for comparison (*Table 1*). By convention, only reference proteins are annotated in current protein databases. As expected, altORFs are on average small, with a size ranging from 30 to 1480 codons. Accordingly, the median size of predicted human alternative proteins is 45 amino acids compared to 460 for reference proteins (*Figure 1c*), and 92.96% of alternative proteins have less than 100 amino acids. Thus, the bulk of the translation products of altORFs would be small proteins. The majority of altORFs either overlap annotated CDSs in a different reading frame (35.98%) or are located in 3'UTRs (40.09%) (*Figure 1d*). 9.83% of altORFs are located in repeat sequences (*Figure 1—figure supplement 1a*), compared to 2.45% of CDSs. To assess whether observed altORFs could be attributable solely to random occurrence, due for instance to the base composition of the transcriptome, we estimated the expected number of altORFs generated in 100 shuffled human transcriptomes. Overall, we observed 62,307 more altORFs than would be expected from random occurrence alone (*Figure 1e*; p<0.0001). This analysis suggests that a large number are expected by chance alone but that at the same time, a large absolute number could potentially be maintained and be functional. The density of altORFs observed in the CDSs, 3'UTRs and ncRNAs (*Figure 1f*) was markedly higher than in the shuffled transcriptomes, suggesting that these are maintained at frequencies higher than expected by chance, again potentially due to their coding function. In contrast, the density of altORFs observed in 5'UTRs was much lower than in the shuffled transcriptomes, supporting recent claims that negative selection eliminates AUGs (and thus the potential for the evolution of altORFs) in these regions (*Iacono et al., 2005*; *Neafsey and Galagan, 2007*).

Although the majority of human annotated CDSs do not have a TIS with a Kozak motif (*Figure 1g*) (*Smith et al., 2005*), there is a correlation between a Kozak motif and translation efficiency (*Pop et al., 2014*). We find that 27,539 (15% of 183,191) human altORFs encoding predicted

**Table 1.** AltORF and alternative protein annotations in different organisms.

| Genomes | Features | | | | | |
|---|---|---|---|---|---|---|
| | Transcripts | | Current annotations | | Annotations of alternative protein coding sequences | |
| | mRNAs | Others1* | CDSs | Proteins | AltORFs | Alternative proteins |
| *H. sapiens* GRCh38 RefSeq GCF_000001405.26 | 67,765 | 11,755 | 68,264 | 54,498 | 539,134 | 183,191 |
| *P. troglodytes* 2.1.4 RefSeq GCF_000001515.6 | 55,034 | 7527 | 55,243 | 41,774 | 416,515 | 161,663 |
| *M. musculus* GRCm38p2, RefSeq GCF_000001635.22 | 73,450 | 18,886 | 73,55 1 | 53,573 | 642,203 | 215,472 |
| *B. Taurus* UMD3.1.86 | 22,089 | 838 | 22,089 | 21,915 | 79,906 | 73,603 |
| *X. tropicalis* Ensembl JGI_4.2 | 28,462 | 4644 | 28,462 | 22,614 | 141,894 | 69,917 |
| *D. rerio* Ensembl ZV10.84 | 44,198 | 8196 | 44,198 | 41,460 | 214,628 | 150,510 |
| *D. melanogaster* RefSeq GCA_000705575.1 | 30,255 | 3474 | 30,715 | 20,995 | 174,771 | 71,705 |
| *C. elegans* WBcel235, RefSeq GCF_000002985.6 | 28,653 | 25,256 | 26,458 | 25,750 | 131,830 | 45,603 |
| *S. cerevisiae* YJM993_v1, RefSeq GCA_000662435.1 | 5471 | 1463 | 5463 | 5423 | 12,401 | 9492 |

*Other transcripts include miRNAs, rRNAs, ncRNAs, snRNAs, snoRNAs, tRNAs.

†Annotated retained-intron and processed transcripts were classified as mRNAs.

DOI: https://doi.org/10.7554/eLife.27860.009

alternative proteins have a Kozak motif (A/GNNAUGG), as compared to 19,745 (36% of 54,498) for annotated CDSs encoding reference proteins (*Figure 1g*). The number of altORFs with Kozak motifs is significantly higher in the human transcriptome compared to shuffled transcriptomes (*Figure 1— figure supplement 2*), again supporting their potential role as protein coding.

## Conservation analyses

Next, we compared evolutionary conservation patterns of altORFs and CDSs. A large number of human alternative proteins have homologs in other species. In mammals, the number of homologous alternative proteins is higher than the number of homologous reference proteins (*Figure 2a*), and nine are even conserved from human to yeast (*Figure 2b*), supporting a potential functional role. As phylogenetic distance from human increases, the number and percentage of genes encoding homologous alternative proteins decreases more rapidly than the percentage of genes encoding reference proteins (*Figure 2a and c*). This observation indicates either that altORFs evolve more rapidly than CDSs or that distant homologies are less likely to be detected given the smaller sizes of alternative proteins. Another possibility is that they evolve following the patterns of evolution of genes that evolve de novo, with a rapid birth and death rate, which accelerates their turnover over time (*Schlötterer, 2015*).

If altORFs play a functional role, they would be expected to be under purifying selection. The first and second positions of a codon experience stronger purifying selection than the third because of redundancy in the genetic code (*Pollard et al., 2010*). In the case of CDS regions overlapping altORFs with a shifted reading frame, the third codon positions of the CDSs are either the first or the second in the altORFs, and should thus also undergo purifying selection. We analyzed conservation of third codon positions of CDSs for 100 vertebrate species for 1088 altORFs completely nested within and conserved across vertebrates (human to zebrafish) with their 889 CDSs from 867 genes (*Figure 3*). We observed that in regions of the CDS overlapping altORFs, third codon positions were evolving at significantly more extreme speeds (slow or quick) than third codon positions of random control sequences from the entire CDS (*Figure 3*), reaching up to 67-fold for conservation at $p<0.0001$ and 124-fold for accelerated evolution at $p<0.0001$. This is illustrated with three altORFs located within the CDS of *NTNG1*, *RET* and *VTI1A* genes (*Figure 4*). These three genes encode a protein promoting neurite outgrowth, the proto-oncogene tyrosine-protein kinase receptor Ret and a protein mediating vesicle transport to the cell surface, respectively. Two of these alternative proteins have been detected by ribosome profiling (*RET*, IP_182668.1) or mass spectrometry (*VTI1A*, IP_188229.1) (see *Supplementary files 1* and *2*).

## Evidence of expression of alternative proteins

We provide two lines of evidence indicating that thousands of altORFs are translated into proteins. First, we re-analyzed detected TISs in publicly available ribosome profiling data (*Michel et al., 2014*; *Raj et al., 2016*), and found 26,531 TISs mapping to annotated CDSs and 12,616 mapping to altORFs in these studies (*Figure 5a*; *Supplementary file 1*). Only a small fraction of TISs detected by ribosomal profiling mapped to altORFs[3'] even if those are more abundant than altORF[5'] relative to shuffled transcriptomes, likely reflecting a recently resolved technical issue which prevented TIS detection in 3'UTRs (*Miettinen and Björklund, 2015*). New methods to analyze ribosome profiling data are being developed and will likely uncover more translated altORFs (*Ji et al., 2015*). In agreement with the presence of functional altORFs[3'], cap-independent translational sequences were recently discovered in human 3'UTRs (*Weingarten-Gabbay et al., 2016*). Second, we re-analyzed proteomic data using our composite database containing alternative proteins in addition to annotated reference proteins (*Figure 5b*; *Supplementary file 2*). We selected four studies representing different experimental paradigms and proteomic applications: large-scale (*Hein et al., 2015*) and targeted (*Tong et al., 2014*) protein/protein interactions, post-translational modifications (*Sharma et al., 2014*), and a combination of bottom-up, shotgun and interactome proteomics (*Rosenberger et al., 2014*). In the first dataset, we detected 3957 predicted alternative proteins in the interactome of reference proteins (*Hein et al., 2015*), providing a framework to uncover the function of these proteins. In a second proteomic dataset containing about 10,000 reference human proteins (*Rosenberger et al., 2014*), a total of 549 predicted alternative proteins were detected. Using a phosphoproteomic large data set (*Sharma et al., 2014*), we detected 384 alternative

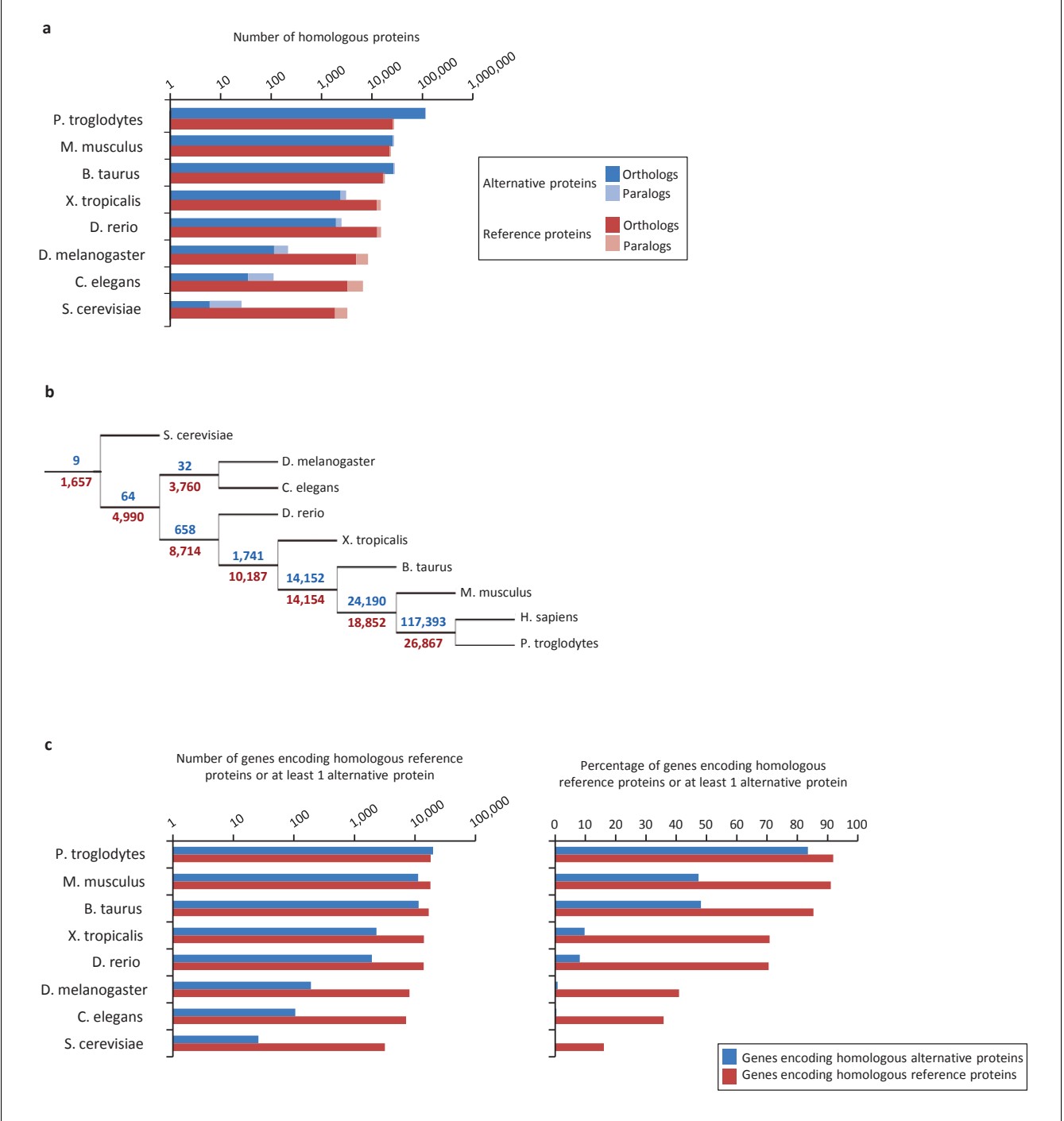

**Figure 2.** Conservation of alternative and reference proteins across different species. (**a**) Number of orthologous and paralogous alternative and reference proteins between *H. sapiens* and other species (pairwise study). (**b**) Phylogenetic tree: conservation of alternative (blue) and reference (red) proteins across various eukaryotic species. (**c**) Number and fraction of genes encoding homologous reference proteins or at least one homologous alternative protein between *H. sapiens* and other species (pairwise study).

DOI: https://doi.org/10.7554/eLife.27860.010

The following source data is available for figure 2:

**Source data 1.** Conservation of alternative and reference proteins across different species.

DOI: https://doi.org/10.7554/eLife.27860.011

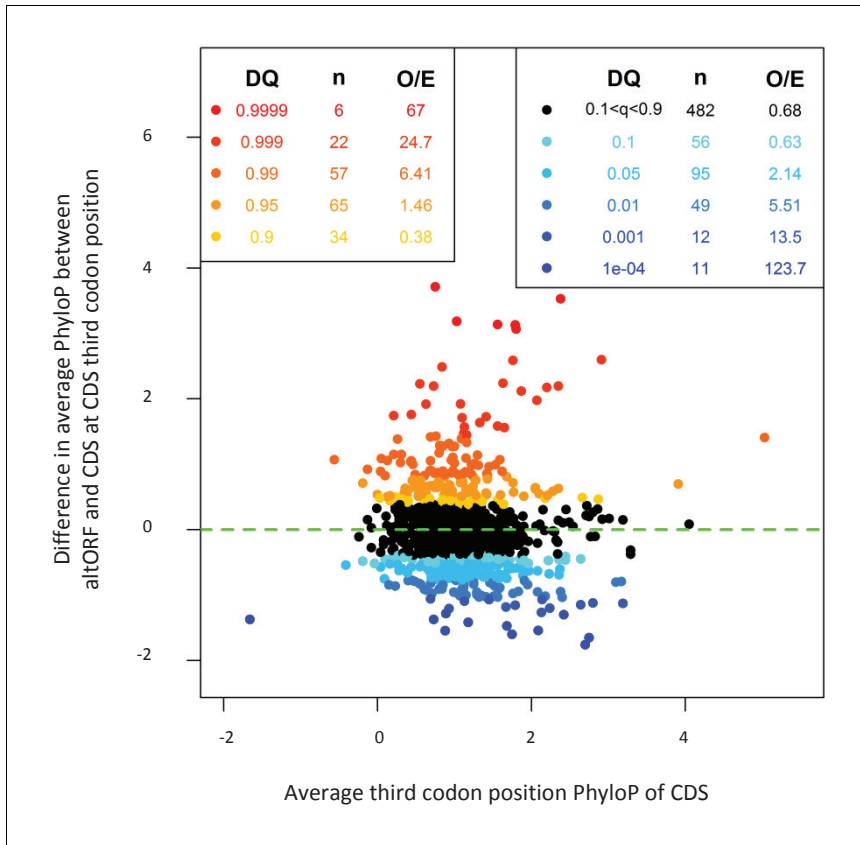

**Figure 3.** AltORFs completely nested within CDSs show more extreme PhyloP values (more conserved or faster evolving) than their CDSs. Differences between altORF and CDS PhyloP scores (altORF PhyloP – CDS PhyloP, *y*-axis) are plotted against PhyloPs for their respective CDSs (*x*-axis). We restricted the analysis to altORF-CDS pairs that were conserved from humans to zebrafish. The plot contains 889 CDSs containing at least one fully nested altORF, paired with one of its altORFs selected at random (to avoid problems with statistical non-independence). PhyloPs for both altORFs and CDSs are based on third codons in the CDS reading frame, calculated across 100 vertebrate species. We compared these differences to those generated based on five random regions in CDSs with a similar length as altORFs. Expected quantiles of the differences ('DQ' columns) were identified and compared to the observed differences. We show the absolute numbers ('n') and observed-to-expected ratios ('O/E') for each quantile. There are clearly substantial over-representations of extreme values (red signaling conservation DQ 0.95, and blue signaling accelerated evolution DQ 0.05) with 317 of 889 altORFs (35.7%). A random distribution would have implied a total of 10% (or 89) of altORFs in the extreme values. This suggests that 25.7% (35.7delete–10%) of these 889 altORFs undergo specific selection different from random regions in their CDSs with a similar length distribution.

DOI: https://doi.org/10.7554/eLife.27860.012

The following source data is available for figure 3:

**Source data 1.** Number of orthologous and co-conserved alternative and reference proteins between H. sapiens and other species (pairwise).

DOI: https://doi.org/10.7554/eLife.27860.013

proteins. The biological function of these proteins is supported by the observation that some alternative proteins are specifically phosphorylated in cells stimulated by the epidermal growth factor, and others are specifically phosphorylated during mitosis (*Figure 6*; *Supplementary file 3*). We provide examples of spectra validation (*Figure 6—figure supplement 1*). A fourth proteomic dataset contained 77 alternative proteins in the epidermal growth factor receptor interactome (*Tong et al., 2014*) (*Figure 5b*). A total of 4872 different alternative proteins were detected in these proteomic data. The majority of these proteins are coded by altORF[CDS], but there are also significant contributions of altORF[3'], altORF[nc] and altORF[5'] (*Figure 5c*). Overall, by mining the proteomic and ribosomal

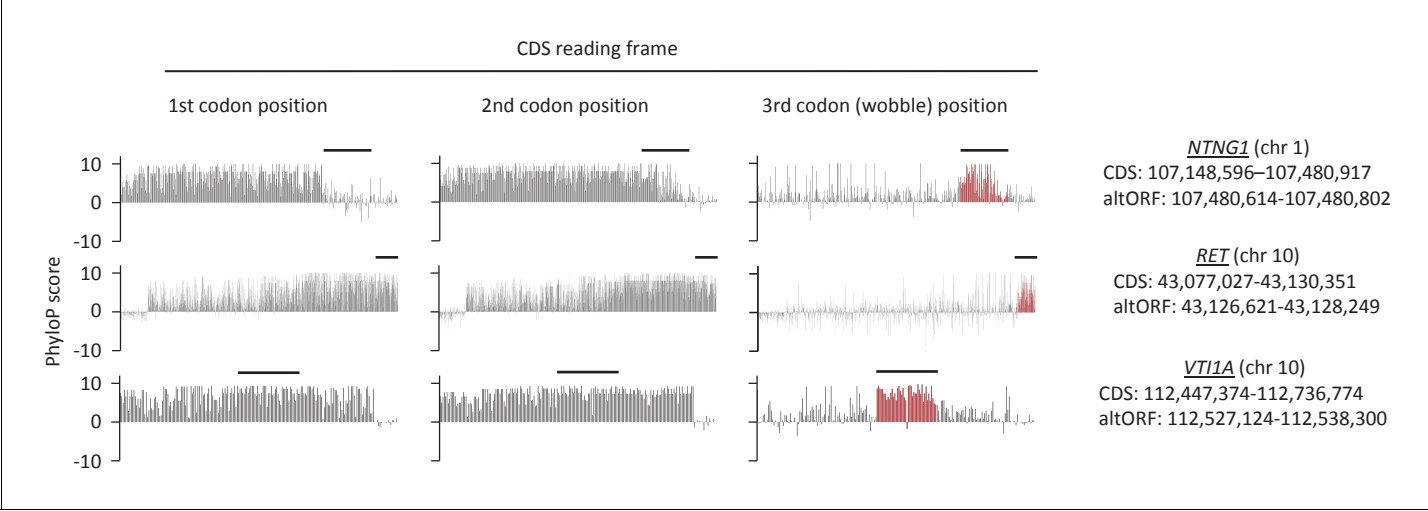

**Figure 4.** First, second, and third codon nucleotide PhyloP scores for 100 vertebrate species for the CDSs of the *NTNG1*, *RET* and *VTI1A* genes. Chromosomal coordinates for the different CDSs and altORFs are indicated on the right. The regions highlighted in red indicate the presence of an altORF characterized by a region with elevated PhyloP scores for wobble nucleotides. The region of the altORF is indicated by a black bar above each graph.

DOI: https://doi.org/10.7554/eLife.27860.014

The following source data is available for figure 4:

**Source data 1.** AltORFs completely nested within CDSs show more extreme PhyloP values (more conserved or faster evolving) than their CDSs.
DOI: https://doi.org/10.7554/eLife.27860.015

profiling data, we detected the translation of a total of 17,371 unique alternative proteins. 467 of these alternative proteins were detected by both MS and ribosome profiling (*Figure 7*), providing a high-confidence collection of small alternative proteins for further studies.

## Functional annotations of alternative proteins

An important goal of this study is to associate potential functions to alternative proteins, which we can do through annotations. Because the sequence similarities and the presence of particular signatures (families, domains, motifs, sites) are a good indicator of a protein's function, we analyzed the sequence of the predicted alternative proteins in several organisms with InterProScan, an analysis and classification tool for characterizing unknown protein sequences by predicting the presence of combined protein signatures from most main domain databases (*Mitchell et al., 2015*) (*Figure 8*; *Figure 8—figure supplement 1*). We found 41,511 (23%) human alternative proteins with at least one InterPro signature (*Figure 8b*). Of these, 37,739 (or 20.6%) are classified as small proteins. Interestingly, the reference proteome has a smaller proportion (840 or 1.68%) of small proteins with at least one InterPro signature, supporting a biological activity for alternative proteins.

Similar to reference proteins, signatures linked to membrane proteins are abundant in the alternative proteome and represent more than 15,000 proteins (*Figure 8c–e*; *Figure 8—figure supplement 1*). With respect to the targeting of proteins to the secretory pathway or to cellular membranes, the main difference between the alternative and the reference proteomes lies in the very low number of proteins with both signal peptides and transmembrane domains. Most of the alternative proteins with a signal peptide do not have a transmembrane segment and are predicted to be secreted (*Figure 8c,d*), supporting the presence of large numbers of alternative proteins in plasma (*Vanderperre et al., 2013*). The majority of predicted alternative proteins with transmembrane domains have a single membrane spanning domain but some display up to 27 transmembrane regions, which is still within the range of reference proteins that show a maximum of 33 (*Figure 8e*).

We extended the functional annotation using the Gene Ontology. A total of 585 alternative proteins were assigned 419 different InterPro entries, and 343 of them were tentatively assigned 192 gene ontology terms (*Figure 9*). 15.5% (91/585) of alternative proteins with an InterPro entry were detected by MS or/and ribosome profiling, compared to 13.7% (22,055/161,110) for alternative

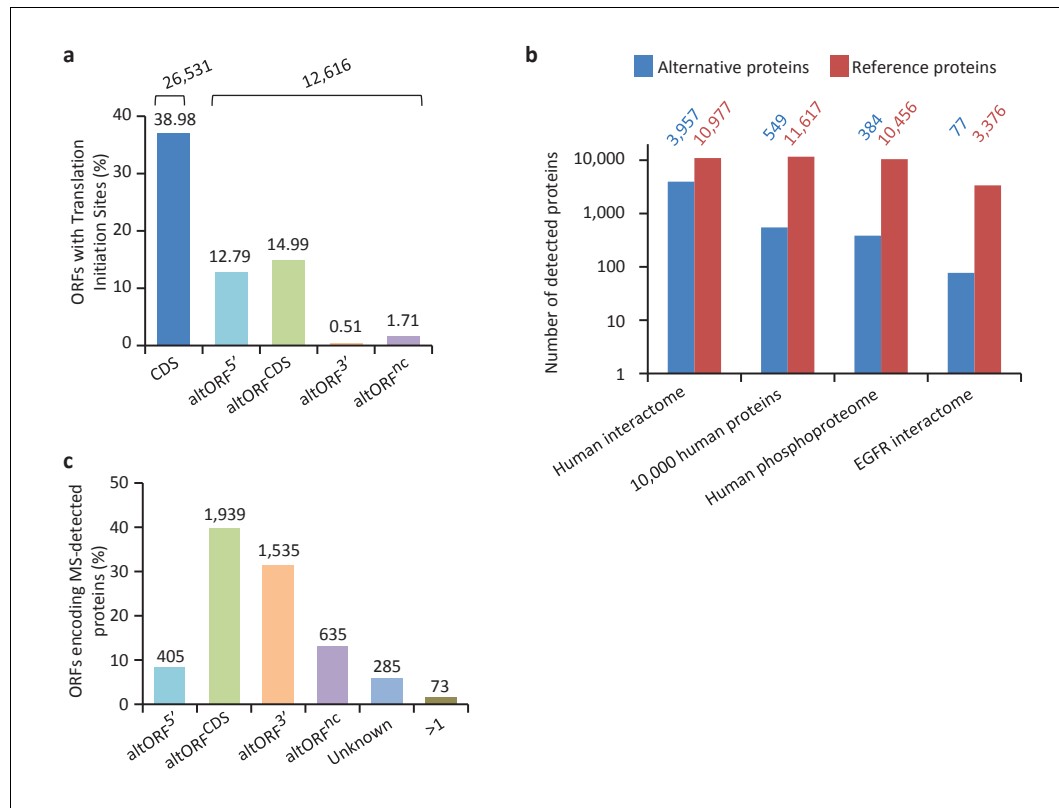

**Figure 5.** Expression of human altORFs. (**a**) Percentage of CDSs and altORFs with detected TISs by ribosomal profiling and footprinting of human cells (**Iacono et al., 2005**). The total number of CDSs and altORFs with a detected TIS is indicated at the top. (**b**) Alternative and reference proteins detected in three large proteomic datasets: human interactome (**Hein et al., 2015**), 10,000 human proteins (**Rosenberger et al., 2014**), human phosphoproteome (**Sharma et al., 2014**), EGFR interactome (**Tong et al., 2014**). Numbers are indicates above each column. (**c**) Percentage of altORFs encoding alternative proteins detected by MS-based proteomics. The total number of altORFs is indicated at the top. Localization 'Unknown' indicates that the detected peptides can match more than one alternative protein. Localization '>1' indicates that the altORF can have more than one localization in different RNA isoforms.

DOI: https://doi.org/10.7554/eLife.27860.016

The following source data and figure supplements are available for figure 5:

**Source data 1.** Expression of human altORFs.
DOI: https://doi.org/10.7554/eLife.27860.021
**Figure supplement 1.** Spectra validation for altSLC35A4[5'].
DOI: https://doi.org/10.7554/eLife.27860.017
**Figure supplement 2.** Spectra validation for altRELT[5'].
DOI: https://doi.org/10.7554/eLife.27860.018
**Figure supplement 3.** Spectra validation for altLINC01420[nc].
DOI: https://doi.org/10.7554/eLife.27860.019
**Figure supplement 4.** Spectra validation for altSRRM2[CDS].
DOI: https://doi.org/10.7554/eLife.27860.020

proteins without an InterPro entry (p-value=1.13e-05, Fisher's exact test and chi-square test). Thus, predicted alternative proteins with InterPro entries are more likely to be detected, supporting their functional role. The most abundant class of predicted alternative proteins with at least one InterPro entry are C2H2 zinc finger proteins with 110 alternative proteins containing 187 C2H2-type/integrase DNA-binding domains, 91 C2H2 domains and 23 C2H2-like domains (**Figure 10a**). Eighteen of these (17.8%) were detected in public proteomic and ribosome profiling datasets (**Table 2**), a percentage that is similar to reference zinc finger proteins (20.1%). Alternative proteins have between 1 and 23 zinc finger domains (**Figure 10b**). Zinc fingers mediate protein-DNA, protein-RNA and

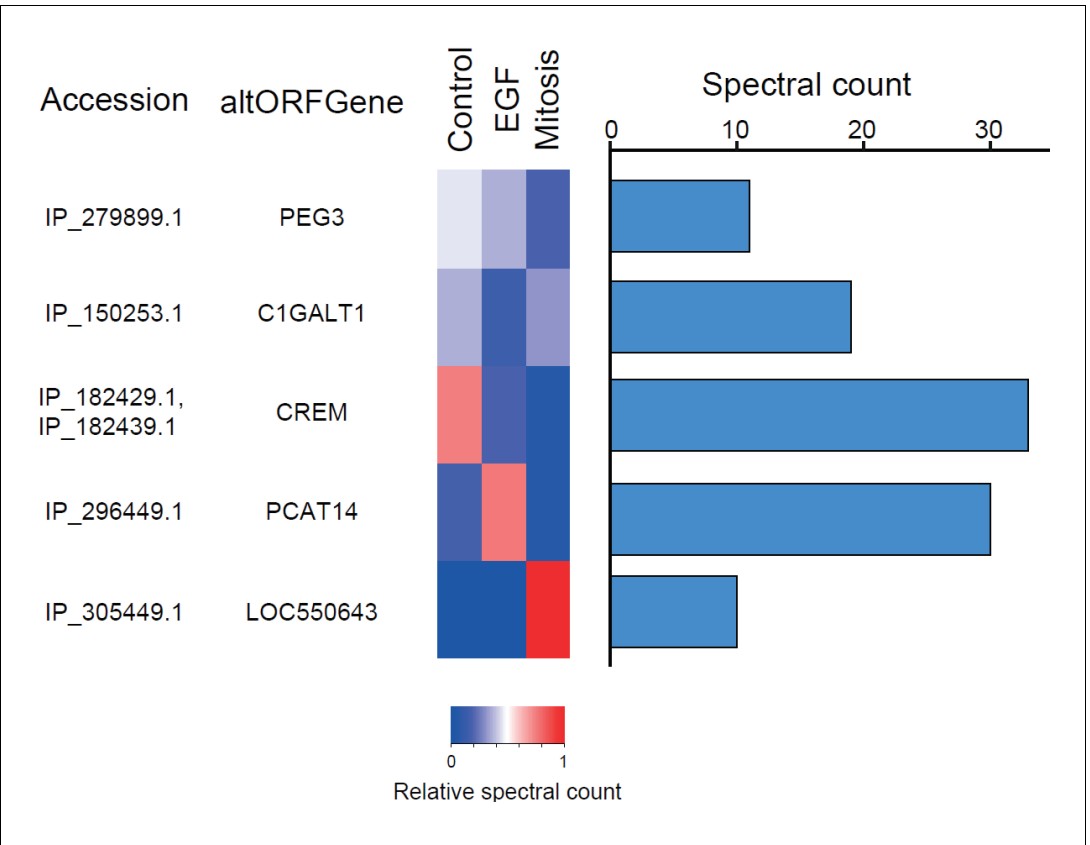

**Figure 6.** The alternative phosphoproteome in mitosis and EGF-treated cells. Heatmap showing relative levels of spectral counts for phosphorylated peptides following the indicated treatment (*Sharma et al., 2014*). For each condition, heatmap colors show the percentage of spectral count on total MS/MS phosphopeptide spectra. Blue bars on the right represent the number of MS/MS spectra; only proteins with spectral counts above 10 are shown.
DOI: https://doi.org/10.7554/eLife.27860.022

The following figure supplement is available for figure 6:

**Figure supplement 1.** Example of a phosphorylated peptide in mitosis - alternative protein AltLINC01420[nc] (LOC550643, IP_305449.1).
DOI: https://doi.org/10.7554/eLife.27860.023

protein-protein interactions (*Wolfe et al., 2000*). The linker sequence separating adjacent finger motifs matches or resembles the consensus TGEK sequence in nearly half the annotated zinc finger proteins (*Laity et al., 2001*). This linker confers high-affinity DNA binding and switches from a flexible to a rigid conformation to stabilize DNA binding. The consensus TGEK linker is present 46 times in 31 alternative zinc finger proteins (*Supplementary file 4*). These analyses show that a number of alternative proteins can be classified into families and will help deciphering their functions.

We compared the functional annotations of the 585 alternative proteins with an InterPro entry with the reference proteins expressed from the same genes. Strikingly, 89 of 110 altORFs coding for zinc finger proteins (*Figure 10*) are present in transcripts in which the CDS also codes for a zinc finger protein. Overall, 138 alternative/reference protein pairs have at least one identical InterPro entry and many pairs have more than one identical entry (*Figure 11a*). The number of identical entries was much higher than expected by chance (*Figure 11b*, p<0.0001). The correspondence between InterPro domains of alternative proteins and their corresponding reference proteins coded by the same genes also indicates that even when entries are not identical, the InterPro terms are functionally related (*Figure 11c*; *Figure 11—figure supplement 1*). The presence of identical domains remains significant (p<0.001) even when the most frequent domains, zinc fingers, are not considered (*Figure 11—figure supplement 2*).

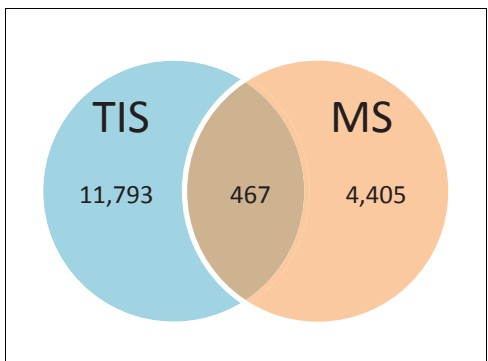

**Figure 7.** Number of alternative proteins detected by ribosome profiling and mass spectrometry. The expression of 467 alternative proteins was detected by both ribosome profiling (translation initiation sites, TIS) and mass spectrometry (MS).

DOI: https://doi.org/10.7554/eLife.27860.024

The following source data is available for figure 7:

**Source data 1.** Number of alternative proteins detected by ribosome profiling and mass spectrometry.

DOI: https://doi.org/10.7554/eLife.27860.025

The presence of identical domains within alternative/reference protein pairs encoded in the same genes may result from alternative splicing events which connect an altORF in a different reading frame than the CDS with a coding exon in the CDS reading frame. Such alternative proteins would likely be unannotated isoforms of the corresponding reference proteins. Thus, we examined whether there is any association between the % of overlap or identity and functional similarity within alternative/reference protein pairs. We performed blast searches of the 183,191 predicted alternative proteins against 54,498 reference proteins using BlastP. All altORFs with more than 80% identity and overlap had already been removed to generate our database (as indicated in the Materials and methods). We found 100 (0.055%) alternative proteins with 25% to 100% identity and 10% to 100% overlap with their reference protein pairs. Among them, 20 (0.00054%) alternative proteins have identical InterPro signatures with their respective reference proteins (**Supplementary file 5**). The distribution of the percentage of sequence identity and overlap between alternative-reference protein pairs with (w/, n = 20) or without (w/o, n = 80) identical Interpro signature is shown in **Figure 11—figure supplement 3**. We observed no significant differences between the two groups (p-value=0.6272; Kolmogorov Smirnov test). We conclude that there is no significant association between identity/overlap and the presence of identical domains in alternative/reference protein pairs.

Recently, the interactome of 118 human zinc finger proteins was determined by affinity purification followed by mass spectrometry (**Schmitges et al., 2016**). This study provides a unique opportunity to test if, in addition to possessing zinc finger domains, some pairs of reference and alternative proteins coded by the same gene may functionally interact. We re-analyzed the MS data using our alternative protein sequence database to detect alternative proteins in this interactome (**Supplementary file 6**). Five alternative proteins (IP_168460.1, IP_168527.1, IP_270697.1, IP_273983.1, IP_279784.1) were identified within the interactome of their reference zinc finger proteins. This number was higher than expected by chance ($p<10^{-6}$) based on 1 million binomial simulations of randomized interactomes. These physical interactions within zinc finger alternative/reference protein pairs suggest that there are examples of functional relationships between large and small proteins coded by the same genes.

## Function of alternative proteins

Finally, we integrated the expression analyses and the conservation analyses of alternative/reference protein pairs to produce a high-confidence list of alternative proteins predicted to have a function and found 2715 alternative proteins in mammals (*H. sapiens* to *B. taurus*), and 44 in vertebrates (*H. sapiens* to *D. rerio*) (**Supplementary file 7**). From this list, we focused on alternative proteins detected with at least two peptide spectrum matches or with high TIS reads and selected altMiD51 (IP_294711.1) among the top 2% of alternative proteins detected with the highest number of unique peptides in proteomics studies, and altDDIT3 (IP_211724.1) among the top 2% of altORFs with the most cumulative reads in translation initiation ribosome profiling studies.

AltMiD51 is a 70 amino acid alternative protein conserved in vertebrates (**Andreev et al., 2015**) and conserved with its reference protein MiD51 from humans to zebrafish (**Supplementary file 7**). Its coding sequence is present in exon 2 of the *MiD51/MIEF1/SMCR7L* gene. This exon forms part of the 5'UTR for the canonical mRNA and is annotated as non-coding in current gene databases (**Figure 12a**). Yet, altMiD51 is robustly detected by MS in several cell lines (**Supplementary file 2**:

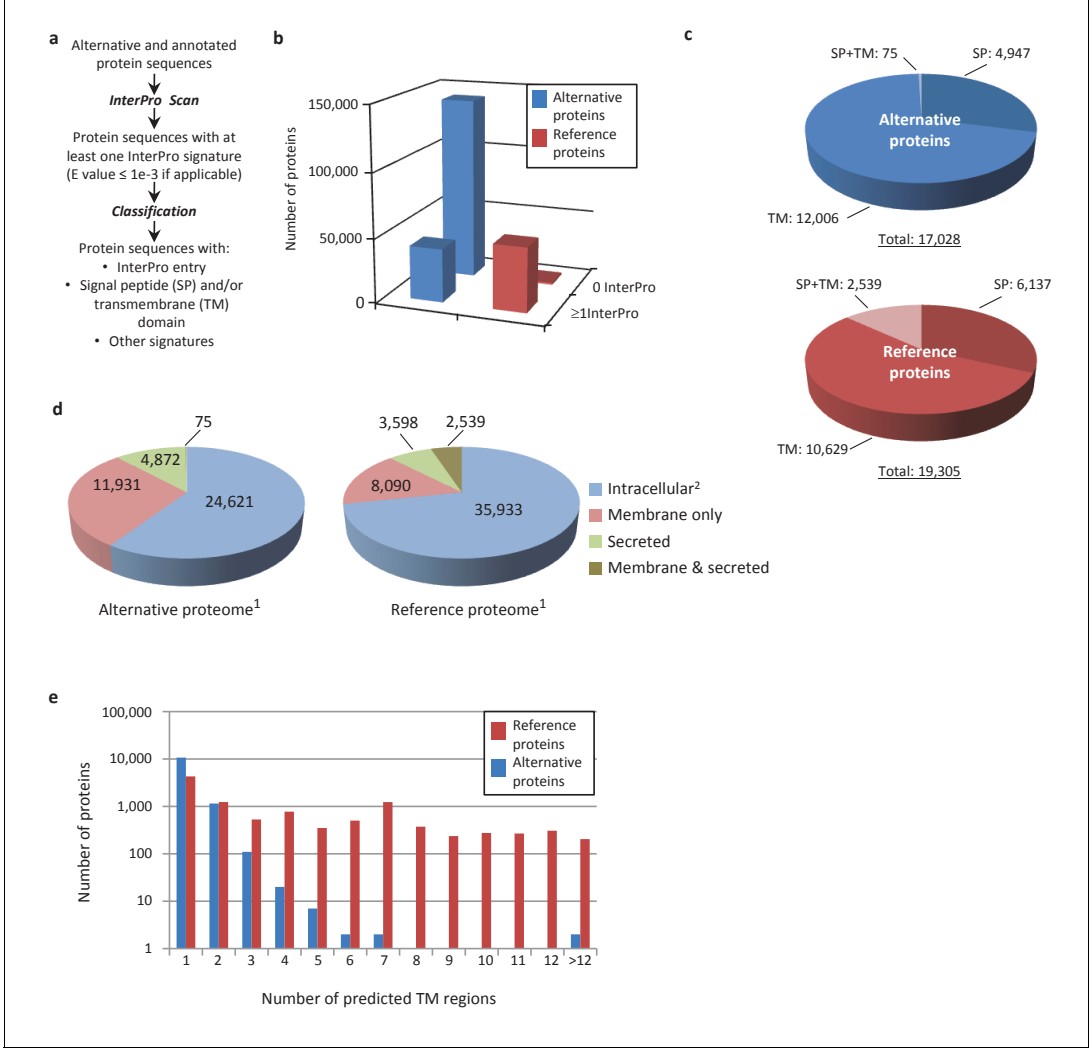

**Figure 8.** Human alternative proteome sequence analysis and classification using InterProScan. (a) InterPro annotation pipeline. (b) Alternative and reference proteins with InterPro signatures. (c) Number of alternative and reference proteins with transmembrane domains (TM), signal peptides (S) and both TM and SP. (d) Number of all alternative and reference proteins predicted to be intracellular, membrane, secreted and membrane-spanning and secreted (*Ingolia et al., 2011*). Proteins with at least one InterPro signature (*Lee et al., 2012*); proteins with no predicted signal peptide or transmembrane features. (e) Number of predicted TM regions for alternative and reference proteins.

DOI: https://doi.org/10.7554/eLife.27860.026

The following source data and figure supplements are available for figure 8:

**Figure supplement 1.** Alternative proteome sequence analysis and classification in *P. troglodytes*, *M. musculus*, *B. Taurus*, *D. melanogaster* and *S. cerevisiae*.

DOI: https://doi.org/10.7554/eLife.27860.027

**Figure supplement 1—source data 1.** Alternative proteome sequence analysis and classification in *P. troglodytes*, *M. musculus*, *B. Taurus*, *D. melanogaster* and *S. cerevisiae*.

DOI: https://doi.org/10.7554/eLife.27860.028

HEK293, HeLa Kyoto, HeLa S3, THP1 cells and gut tissue), and we validated some spectra using synthetic peptides (*Figure 12—figure supplement 1*), and it is also detected by ribosome profiling (*Supplementary file 1*) (*Vanderperre et al., 2013*; *Andreev et al., 2015*; *Kim et al., 2014*). We confirmed co-expression of altMiD51 and MiD51 from the same transcript (*Figure 12b*). Importantly, the tripeptide LYR motif predicted with InterProScan and located in the N-terminal domain of alt-MiD51 (*Figure 12a*) is a signature of mitochondrial proteins localized in the mitochondrial matrix (*Angerer, 2015*). Since *MiD51/MIEF1/SMCR7L* encodes the mitochondrial protein MiD51, which

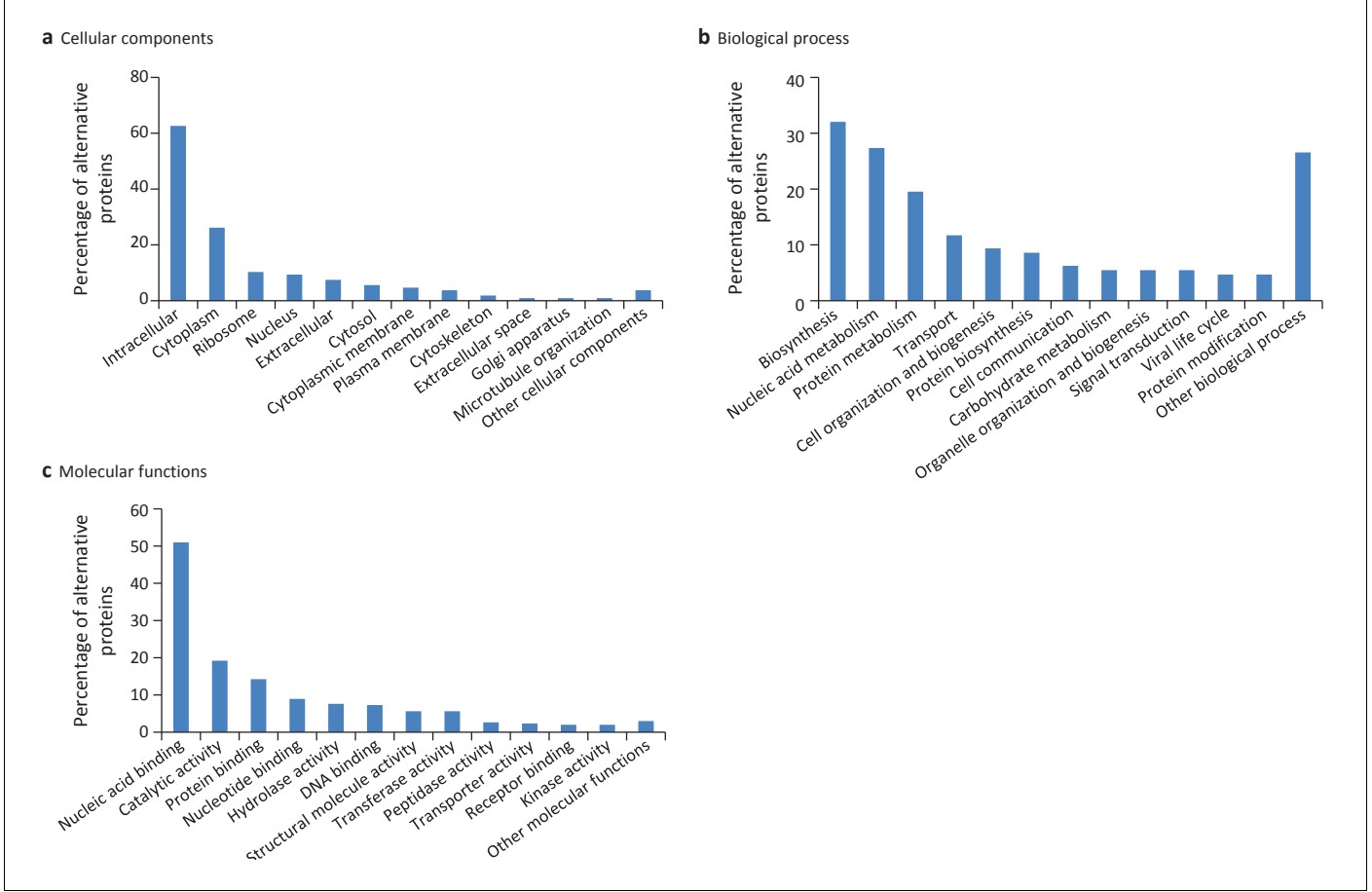

**Figure 9.** Gene ontology (GO) annotations for human alternative proteins. GO terms assigned to InterPro entries are grouped into 13 categories for each of the three ontologies. (**a**) 34 GO terms were categorized into cellular component for 107 alternative proteins. (**b**) 64 GO terms were categorized into biological process for 128 alternative proteins. (**c**) 94 GO terms were categorized into molecular function for 302 alternative proteins. The majority of alternative proteins with GO terms are predicted to be intracellular, to function in nucleic acid-binding, catalytic activity and protein binding and to be involved in biosynthesis and nucleic acid metabolism processes.

DOI: https://doi.org/10.7554/eLife.27860.029

The following source data is available for figure 9:

**Source data 1.** Gene ontology(GO) annotations for human alternative proteins.

DOI: https://doi.org/10.7554/eLife.27860.030

promotes mitochondrial fission by recruiting cytosolic Drp1, a member of the dynamin family of large GTPases, to mitochondria (*Losón et al., 2013*), we tested for a possible functional connection between these two proteins expressed from the same mRNA. We first confirmed that MiD51 induces mitochondrial fission (*Figure 12—figure supplement 2*). Remarkably, we found that altMiD51 also localizes at the mitochondria (*Figure 12c*; *Figure 12—figure supplement 3*) and that its overexpression results in mitochondrial fission (*Figure 12d*). This activity is unlikely to be through perturbation of oxidative phosphorylation since the overexpression of altMiD51 did not change oxygen consumption nor ATP and reactive oxygen species production (*Figure 12—figure supplement 4*). The decrease in spare respiratory capacity in altMiD51-expressing cells (*Figure 12—figure supplement 4a*) likely resulted from mitochondrial fission (*Motori et al., 2013*). The LYR domain is essential for altMiD51-induced mitochondrial fission since a mutant of the LYR domain, altMiD51(LYR→AAA) was unable to convert the mitochondrial morphology from tubular to fragmented (*Figure 12d*). Drp1(K38A), a dominant negative mutant of Drp1 (*Smirnova et al., 1998*), largely prevented the ability of altMiD51 to induce mitochondrial fragmentation (*Figure 12d*; *Figure 12—figure supplement 5a*). In a control experiment, co-expression of wild-type Drp1 and altMiD51 proteins resulted

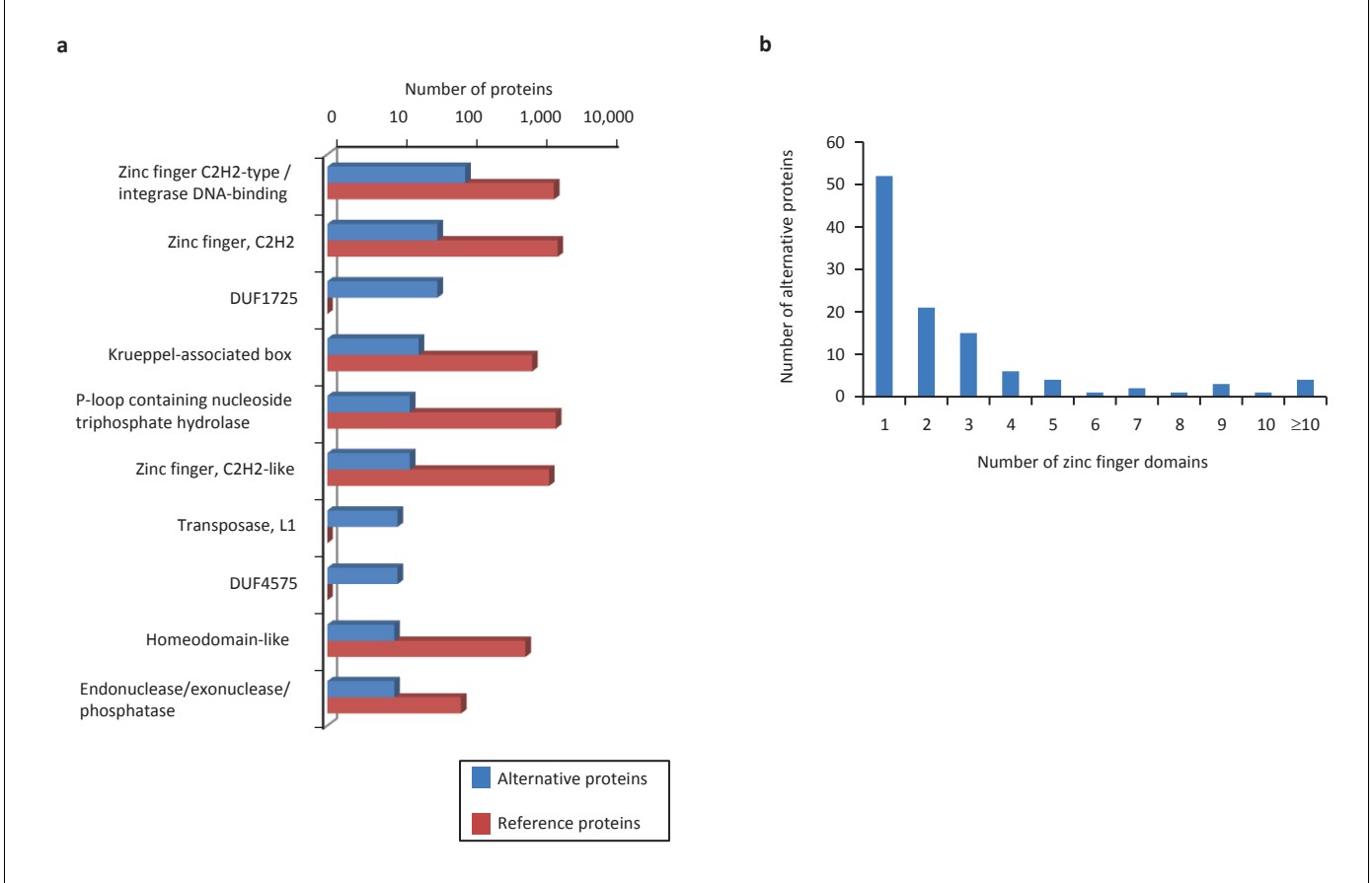

**Figure 10.** Main InterPro entries in human alternative proteins. (**a**) The top 10 InterPro families in the human alternative proteome. (**b**) A total of 110 alternative proteins have between 1 and 23 zinc finger domains.

DOI: https://doi.org/10.7554/eLife.27860.031

The following source data is available for figure 10:

**Source data 1.** Main InterPro entries in human alternative proteins.

DOI: https://doi.org/10.7554/eLife.27860.032

in mitochondrial fragmentation (*Figure 12—figure supplement 5b*). Expression of the different constructs used in these experiments was verified by western blot (*Figure 12—figure supplement 6*). Drp1 knockdown interfered with altMiD51-induced mitochondrial fragmentation (*Figure 13*), confirming the proposition that Drp1 mediates altMiD51-induced mitochondrial fragmentation. It remains possible that altMiD51 promotes mitochondrial fission independently of Drp1 and is able to reverse the hyperfusion induced by Drp1 inactivation. However, Drp1 is the key player mediating mitochondrial fission and most likely mediates altMiD51-induced mitochondrial fragmentation, as indicated by our results.

AltDDIT3 is a 31 amino acid alternative protein conserved in vertebrates and conserved with its reference protein DDIT3 from human to bovine (*Supplementary file 7*). Its coding sequence overlaps the end of exon 1 and the beginning of exon 2 of the *DDIT3/CHOP/GADD153* gene. These exons form part of the 5′UTR for the canonical mRNA (*Figure 14a*). To determine the cellular localization of altDDIT3 and its possible relationship with DDIT3, confocal microscopy analyses were performed on HeLa cells co-transfected with altDDIT3$^{GFP}$ and DDIT3$^{mCherry}$. Expression of these constructs was verified by western blot (*Figure 14—figure supplement 1*). Interestingly, both proteins were mainly localized in the nucleus and partially localized in the cytoplasm (*Figure 14b*). This distribution for DDIT3 confirms previous studies (*Cui et al., 2000*; *Chiribau et al., 2010*). Both proteins seemed to co-localize in these two compartments (Pearson correlation coefficient 0.92, *Figure 14c*). We further confirmed the statistical significance of this colocalization by applying

**Table 2.** Alternative zinc finger proteins detected by mass spectrometry (MS) and ribosome profiling (RP)

| Alternative protein accession | Detection method* | Gene | Amino acid sequence | AltORF localization |
|---|---|---|---|---|
| IP_238718.1 | MS | RP11 | MLVEVACSSCRSLLHKGAGASEDGAALEPAHTGGKENGATT | nc |
| IP_278905.1 | RP | ZNF761 | MSVARPLVGSHILYAIIDFILERNLISVMSVARTLVRSHPLYATIDFILERNLTSVMSVARPLVRSQTLHAIVDFILEKNKCNECGEVFNQQAHLAGHHRIHTGEKP | CDS |
| IP_278745.1 | MS and RP | ZNF816 | MSVARPSVRNHPFNAIIYFTLERNLTNVKNVTMFTFADHTLKDIGRFILERDHTNVRFVTRFSGVIHTLQNIREFILERNHTSVINVAGVSVGSHPFNTIIHFTLERNLTHVMNVARFLVEEKTLHVIIDFMLERNLTNVKNVTKFSVADHTLKDIGEFILGKNHTNVRFVTRLSGVIHALQTIREFILERNLTSVINVRRFLIKKESLHNIREFILERNLTSVMNVARFLIKKQALQNIREFILQRNLTSVMSVAKPLLDSQHLFTIKQSMGVGKLYKCNDCHKVFSNATTIANHYRIHIEERSTSVINVANFSDVIHNL | CDS |
| IP_138289.1 | MS | ZSCAN31 | MNIGGATLERNPINVRSVGKPSVPAMASLDTEESTQGKNHMNAKCVGRLSSSAHALFSIRGYTLERSAISVVSVAKPSFRMQGFSSISESTLVRNPISAVSAVNSLVSGHFLRNIRKSTLERDHKGDEFGKAFSHHCNLIRHFRIHTVPAELD | CDS |
| IP_278564.1 | MS | ZNF808 | MIVTKSSVTLQQLQIIGESMMKRNLLSVINVACFSDIVHTLQFIGNLILERNLTNVMIEARSSVKLHPMQNRRIHTGEKPHKCDDCGKAFTSHSHLVGHQRIHTGQKSCKCHQCGKVFSPRSLLAEHEKIHF | 3'UTR |
| IP_275012.1 | MS | ZNF780A | MKPCECTECGKTFSCSSNIVQHVKIHTGEKRYNVRNMGKHLLWMISCLNIRKFRIVRNFVTIRSVDKPSLCTKNLLNTRELILMRNLVNIKECVKNFHHGLGFAQLLSIHTSEKSLSVRNVGRFIATLNTLEFGEDNSCEKVFE | 3'UTR |
| IP_270595.1† | MS | ZNF440 | MHSVERPYKCKICGRGFYSAKSFQIHEKSYTGEKPYECKQCGKAFVSFTSFRYHERTHTGENPYECKQFGKAFRSVKNLRFHKRTHTGEKPCECKKCRKAFHNFSSLQIHERMHRGEKLCECKHCGKAFISAKIL | CDS |
| IP_270643.1† | MS | ZNF763 | MKKLTLERNPINACHVVKPSIFPVPFSIMKGLTLERNPMSVSVGKPSDVPHTFEGMVGLTGEKPYECKECGKAFRSASHLQIHERTQTHIRIHSGERPYKCKTCGKGFYSPTSFQRHEKTHTAEKPYECKQCGKAFSSSSSFWYHERTHTGEKPYECKQCGKAFRSASIQMHAGTHPEEKPYECKQCGKAFRSAPHLRIHGRTHTGEKPYECKECGKAFRSAKNLRIHERTQTHVRMHSVERPYKCKICGKGFYSAKSFQIPEKSYTGEKPYECKQCGKAFISFTSFR | 3'UTR |
| IP_270597.1‡ | MS | ZNF440 | MKNLTLERNPMSVSNVGKPLFPSLPFDIMKGLTLERTPMSVSNLGKPSDLSKIFDFIKGHTLERNPVNVRNVEKHSIISLLCKYMKGCTEERSSVNVSIVGKHSYLPRSFEYMQEHTMERNPMNVKNAEKHSACLLPFIDMKRLTLEGNTMNASNVAKLSLLPVLFNIMKEHTREKPYQCKQCAKAFISSTSFQYHERTHMGEKPYECMPSGKAFISSSSLQYHERTHTGEKPYEYKQCGKAFRSASHLQMHGRTHTGEKPYECKQYGKAFRPDKIL | 3'UTR |
| IP_270609.1‡ | MS | ZNF439 | MNVSNVAKAFTSSSSFQYHERTHTGEKPYQCKQCGKAVRSASRLQMHGSTHTWQKLYECKQYGKAFRSARIL | 3'UTR |
| IP_270663.1‡ | MS | ZNF844 | MHGRTHTQEKPYECKQCGKAFIFSTSFRYHERTHTGEKPYECKQCGKAFRSATQLQMHRKIHTGEKPYECKQCGKAYRSVSQLLVHERTHTVEQPYEYKQYGKAFRFAKNLQIQTMNVNN | CDS |
| IP_270665.1‡ | MS | ZNF844 | MHRKIHTGEKPYECKQCGKAYRSVSQLLVHERTHTVEQPYEYKQYGKAFRFAKNLQIQTMNVNN | CDS |
| IP_270668.1‡ | MS | ZNF844 | MSSTAFQYHEKTHTREKHYECKQCGKAFISSGSLRYHERTHTGEKPYECKQCGKAFRSATQLQMHRKIHTGEKPYECKQCGKAYRSVSQLLVHERTHTVEQPYEYKQYGKAFRFAKNLQIQTMNVNN | 3'UTR |
| IP_138139.1 | MS | ZNF322 | MLSPSRCKRIHTGEQLFKCLQCQLCCRQYEHLIGPQKTHPGEKPQQV | 3'UTR |
| IP_204754.1 | RP | ZFP91-CNTF | MPGETEEPRPPEQQDQEGGEAAKAAPEEPQQRPPEAVAAAPAGTTSSRVLRGGRDRGRAAAAAAAAAVSRRRKAEYPRRRRSSPSARPPDVPGQQPQAAKSPSPVQGKKSPRLLCIEKVTTDKDPKEEKEEEDDSALPQEVSIAASRPSRGWRSSRTSVSRHRDTENTRSSRSKTGSLQLICKSEPNTDQLDYDVGEEHQSPGGISSEEEEEEEEEMLISEEEIPFKDDPRDETYKPHLERETPKPRRKSGKVKEEKEKKEIKVEVEVEVKEEENEIREDEEPPRKRGRRRKDDKSPRLPKRRKKPPIQYVRCEMEGCGTVLAHPRYLQHHIKYQHLLKKKYVCPHPSCGRLFRLQKQLLRHAKHHTDQRDYICEYCARAFKSSHNLAVHRMIHTGEKPLQCEICGFTCRQKASLNWHMKKHDADSFYQFSCNICGKKFEKKDSVVAHKAKSHPEVLIAEEALAANAGALITSTDILGTNPESLTQPSDGQGLPLLPEPLGNSTSGECLLLEAEGMSKSYCSGTERSIHR | nc |
| IP_098649.1 | RP | INO80B-WBP1 | MSKLWRRGSTSGAMEAPEPGEALELSLAGAHGHGVHKKKHKKHKKKKHKKKHHQEEDAGPTQPSPAKPQLKLKIKLGGQVLGTKSVPTFTVIPEGPRSPSPLMVVDNEEEPMEGVPLEQYRAWLDEDSNLSPSPLRDLSGGLGGQEEEEEQRWLDALEKGELDDNGDLKKEINERLLTARQRALLQKARSQPSPMLPLPVAEGCPPPALTEEMLLKREERARKRRLQAARRAEEHKNQTIERLTKTAATSGRGGRGGARGERRGGRAAAPAPMVRYCSGAQGSTLSFPPGVPAPTAVSQRPSPSGPPPRCSVPGCPHPRRYACSRTGQALCSLQCYRINLQMRLGGPEGPGSPLLATFESCAQE | nc |
| IP_115174.1 | RP | ZNF721 | MYIGEFILERNPTHVENVAKPLDSLQIFMRIRKFILERNPTRVETVAKPLDSLQIFMHIRKFILEIKPYKCKECGKAFKSYYSILKHKRTHTRGMSYEGDECRGL | CDS |

*Table 2 continued on next page*

*Table 2 continued*

| Alternative protein accession | Detection method* | Gene | Amino acid sequence | AltORF localization |
|---|---|---|---|---|
| IP_275016.1 | RP | ZNF780A | MNVRSVGKALIVVHTLFSIRKFIPMRNLLYVGNVRWPLDIIANLLNILEFILVTSHLNVKTVGRPSIVA QALFNIRVFTLVRSPMNVRSVGRLLDFTYNFPNIRKLTQVKNHLNVRNVGNSFVVVQILINIEVFILE RNPLNVRNVGKPFDFICTLFDIRNCILVRNPLNVRSVGKPFDFICNLFDIRNCILVRNPLNVRNVERF LVFPPSLIAIRTFTQVRRHLECKECGKSFNRVSNHVQHQSIRAGVKPCECKGCGKGFICGSNVIQH QKIHSSEKLFVCKEWRTTFRYHYHLFNITKFTLVKNPLNVKNVERPSVF | CDS or 3'UTR |
| IP_278870.1 | RP | ZNF845 | MNVARFLIEKQNLHVIIEFILERNIRNMKNVTKFTVVNQVLKDRRIHTGEKAYKCKSL | CDS |
| IP_278888.1 | RP | ZNF765 | MSVARPSAGRHPLHTIIDFILDRNLTNVKIVMKLSVSNQTLKDIGEFILERNYTCNECGKTFNQE LTLTCHRRLHSGEKPYKYEELDKAYNFKSNLEIHQKIRTEENLTSVMSVARP | CDS |
| IP_278918.1 | RP | ZNF813 | MNVARVLIGKHTLHVIIDFILERNLTSVMNVARFLIEKHTLHIIIDFILEINLTSVMNVARFLIKKHTLH VTIDFILERNLTSVMNVARFLIKKQTLHVIIDFILERNLTSLMSVAKLLIEKQSLHIIIQFILERNKC NECGKTFCHNSVLVIHKNSYWRETSVMNVAKFLINKHTFHVIIDFIVERNLRNVKHVTKFTVANRA SKDRRIHTGEKAYKGEEYHRVFSHKSNLERHKINHTAEKP | CDS |
| IP_280349.1 | RP | ZNF587 | MNAVNVGNHFFPALRFMFIKEFILDKSLISAVNVENPFLNVPVSLNTGEFTLEKGLMNAPNVEKHF SEALPSFIIRVHTGERPYECSEYGKSFAEASRLVKHRRVHTGERPYECCQCGKHQNVCCPRS | CDS |
| IP_280385.1 | RP | ZNF417 | MNAMNVGNHFFPALRFMFIKEFILDKSLISAVNVENPLLNVPVSLNTGEFTLEKGLMNVPNVEKHF SEALPSFIIRVHTGERPYECSEYGKSFAETSRLIKHRRVHTGERPYECCQSGKHQNVCSPWS | CDS |

*MS, mass spectrometry; RP, ribosome profiling.

† These two proteins were not detected with unique peptides but with shared peptides. One protein only was counted in subsequent analyses.

These five proteins were not detected with unique peptides but with shared peptides. One protein only was counted in subsequent analyses.

DOI: https://doi.org/10.7554/eLife.27860.033

Costes' automatic threshold and Costes' randomization colocalization analysis and Manders Correlation Coefficient (*Figure 14d*; *Figure 14—figure supplement 2*) (*Bolte and Cordelières, 2006*). Finally, in lysates from cells co-expressing altDDIT3^GFP and DDIT3^mCherry, DDIT3^mCherry was immunoprecipitated with GFP-trap agarose, confirming an interaction between the small altDDTI3 and the large DDIT3 proteins encoded in the same gene (*Figure 14e*).

## Discussion

We have provided the first functional annotation of altORFs with a minimum size of 30 codons in different genomes. The comprehensive annotation of *H. sapiens* altORFs is freely available to download at https://www.roucoulab.com/p/downloads (*Homo sapiens* functional annotation of alternative proteins based on RefSeq GRCh38 (hg38) predictions). In light of the increasing evidence from approaches such as ribosome profiling and MS-based proteomics that the one mRNA-one canonical CDS assumption is untenable, our findings provide the first clear functional insight into a new layer of regulation in genome function. While many observed altORFs may be evolutionary accidents with no functional role, several independent lines of evidence support translation and a functional role for thousands of alternative proteins: (1) overrepresentation of altORFs relative to shuffled sequences; (2) overrepresentation of altORF Kozak sequences; (3) active altORF translation detected via ribosomal profiling; (4) detection of thousands of alternative proteins in multiple existing proteomic datasets; (5) correlated altORF-CDS conservation, but with overrepresentation of highly conserved and fast-evolving altORFs; (6) overrepresentation of identical InterPro signatures between alternative and reference proteins encoded in the same mRNAs; and (7) presence of clear, striking examples in altMiD51, altDDIT3 and 5 alternative proteins interacting with their reference zinc finger proteins. While far from proven in our study, three of these lines of evidence (5, 6, and 7) would support the intriguing hypothesis that many altORFs code for proteins that cooperate functionally with the proteins coded by their CDSs. This hypothesis would also agree with recently increasing evidence that small proteins often regulate the function of larger proteins (*Couso and Patraquim, 2017*). Further experimental examples and more detailed co-conservation studies will be needed to address this hypothesis.

The presence of two or more coding sequences in the same gene provides a mechanism for coordinated transcriptional regulation. Consequently, the transcription of these coding sequences can be turned on or off together, similar to prokaryotic operons. We speculate that having genes composed

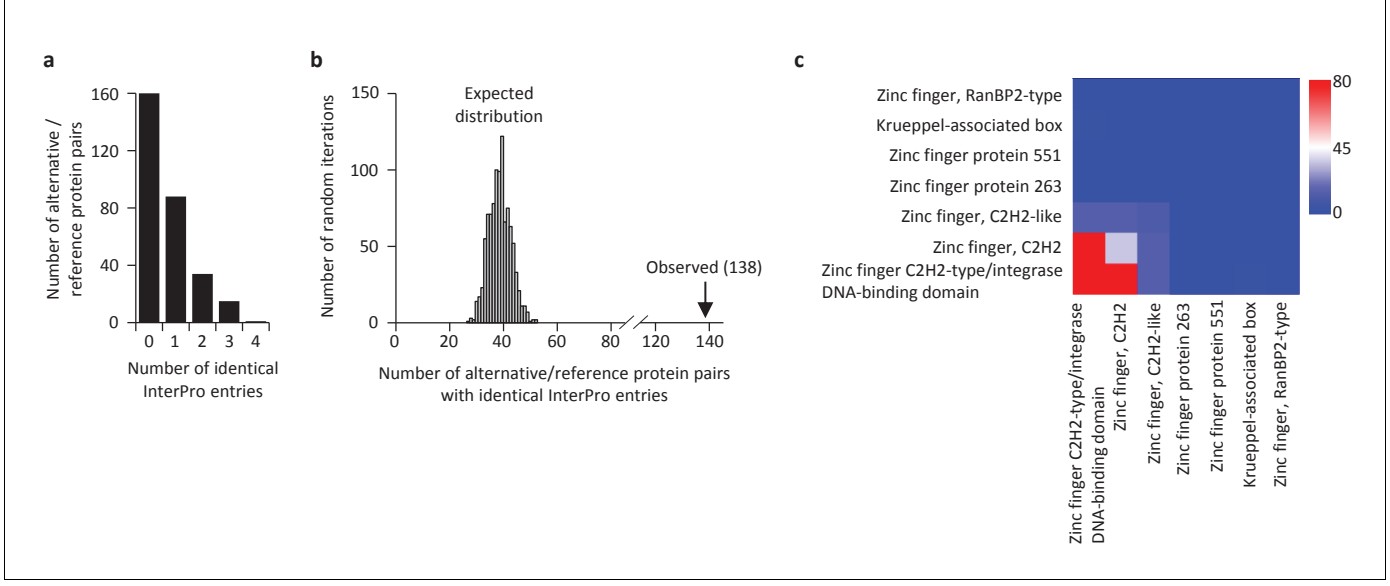

**Figure 11.** Some reference and alternative proteins have identical functional domains. (**a**) Distribution of the number of identical InterPro entries co-ocurring between alternative and reference proteins coded by the same transcripts. 138 pairs of alternative and reference proteins have between 1 and 4 identical protein domains (InterPro entries). Only alternative/reference protein pairs that have at least one identical domain are considered (n = 298). (**b**) The number of reference/alternative protein pairs with identical domains (n = 138) is higher than expected by chance alone. The distribution of expected pairs having identical domains and the observed number are shown. (**c**) Matrix of co-occurrence of domains related to zinc fingers. The entries correspond to the number of times entries co-occur in reference and alternative proteins. The full matrix is available in *Figure 11—figure supplement 1*.

DOI: https://doi.org/10.7554/eLife.27860.034

The following source data and figure supplements are available for figure 11:

**Source data 1.** Distribution of the percentage of sequence identity and overlap between alternative-reference protein pairs with (20) or without (80) identical Interpro signature.

DOI: https://doi.org/10.7554/eLife.27860.038

**Figure supplement 1.** Matrix of co-occurrence of InterPro entries between alternative/reference protein pairs coded by the same transcript.

DOI: https://doi.org/10.7554/eLife.27860.035

**Figure supplement 2.** Some reference and alternative proteins have identical functional domains.

DOI: https://doi.org/10.7554/eLife.27860.036

**Figure supplement 3.** Distribution of the percentage of sequence identity and overlap between alternative-reference protein pairs with (20) or without (80) identical Interpro signature.

DOI: https://doi.org/10.7554/eLife.27860.037

of a CDS and one or more altORFs gives rise to fewer, denser transcription units and allows cells to adapt more quickly with optimized energy expenditure to environmental changes.

Upstream ORFs, here labeled altORFs[5'], are important translational regulators of canonical CDSs in vertebrates (*Johnstone et al., 2016*). Interestingly, the altORF[5'] encoding altDDIT3 was characterized as an inhibitory upstream ORF (*Jousse et al., 2001*; *Young et al., 2016*), but evidence of endogenous expression of the corresponding small protein was not sought. The detection of alt-MiD51 and altDDIT3 suggests that a fraction of altORFs[5'] may have dual functions as translation regulators and functional proteins.

Our results raise the question of the evolutionary origins of these altORFs. A first possible mechanism involves the polymorphism of initiation and stop codons during evolution (*Lee and Reinhardt, 2012*; *Andreatta et al., 2015*). For instance, the generation of an early stop codon in the 5'end of a CDS could be followed by the evolution of another translation initiation site downstream, creating a new independent ORF in the 3'UTR of the canonical gene. This mechanism of altORF origin, reminiscent of gene fission, would at the same time produce a new altORF that has identical protein domains with the annotated CDS, as we observed for a substantial fraction (24%) of the 585 alternative proteins with an InterPro entry. A second mechanism would be de novo origin of ORFs, which

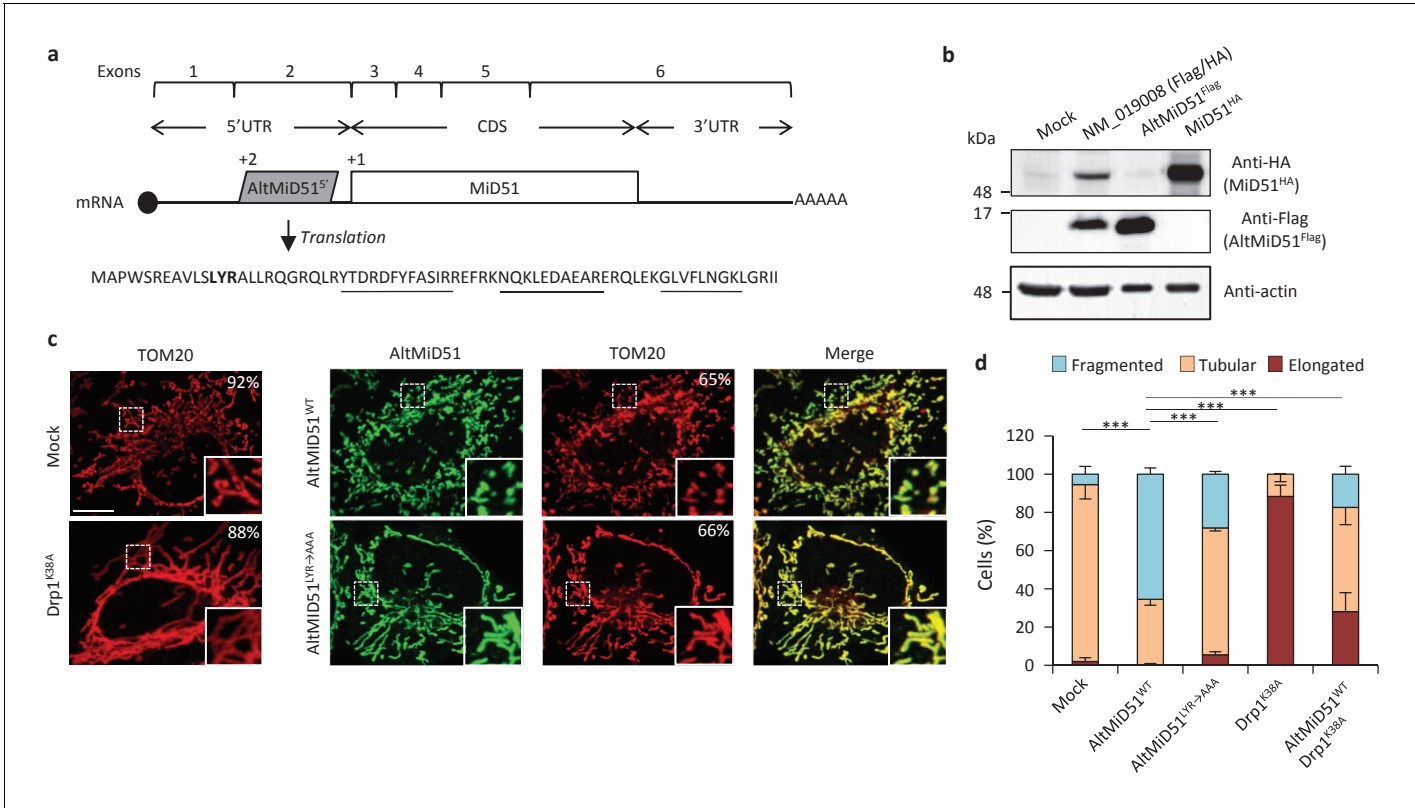

**Figure 12.** AltMiD51[5'] expression induces mitochondrial fission. (**a**) AltMiD51[5'] coding sequence is located in exon two or the *MiD51/MIEF1/SMCR7L* gene and in the 5'UTR of the canonical mRNA (RefSeq NM_019008).+2 and+1 indicate reading frames. AltMiD51 amino acid sequence is shown with the LYR tripeptide shown in bold. Underlined peptides were detected by MS. (**b**) Human HeLa cells transfected with empty vector (mock), a cDNA corresponding to the canonical MiD51 transcript with a Flag tag in frame with altMiD51 and an HA tag in frame with MiD51, altMiD51[Flag] cDNA or MiD51[HA] cDNA were lysed and analyzed by western blot with antibodies against Flag, HA or actin, as indicated. (**c**) Confocal microscopy of mock-transfected cells, cells transfected with altMiD51[WT], altMiD51[LYR→AAA] or Drp1[K38A] immunostained with anti-TOM20 (red channel) and anti-Flag (green channel) monoclonal antibodies. In each image, boxed areas are shown at higher magnification in the bottom right corner. % of cells with the most frequent morphology is indicated: mock (tubular), altMiD51[WT] (fragmented), altMiD51(LYR→AAA) (tubular), Drp1(K38A) (elongated). Scale bar, 10 mm. (**d**) Bar graphs show mitochondrial morphologies in HeLa cells. Means of three independent experiments per condition are shown (100 cells for each independent experiment). ***p<0.0005 (Fisher's exact test) for the three morphologies between altMiD51(WT) and the other experimental conditions.
DOI: https://doi.org/10.7554/eLife.27860.039

The following source data and figure supplements are available for figure 12:

**Source data 1.** Mitochondrial morphologies in HeLa cells.
DOI: https://doi.org/10.7554/eLife.27860.046
**Figure supplement 1.** Spectra validation for altMiD51.
DOI: https://doi.org/10.7554/eLife.27860.040
**Figure supplement 2.** MiD51 expression results in mitochondrial fission.
DOI: https://doi.org/10.7554/eLife.27860.041
**Figure supplement 3.** AltMiD51 is localized in the mitochondrial matrix.
DOI: https://doi.org/10.7554/eLife.27860.042
**Figure supplement 4.** Mitochondrial function parameters.
DOI: https://doi.org/10.7554/eLife.27860.043
**Figure supplement 5.** Representative confocal images of cells co-expressing altMiD51[GFP] and Drp1(K38A)[HA].
DOI: https://doi.org/10.7554/eLife.27860.044
**Figure supplement 6.** Protein immunoblot showing the expression of different constructs in HeLa cells.
DOI: https://doi.org/10.7554/eLife.27860.045

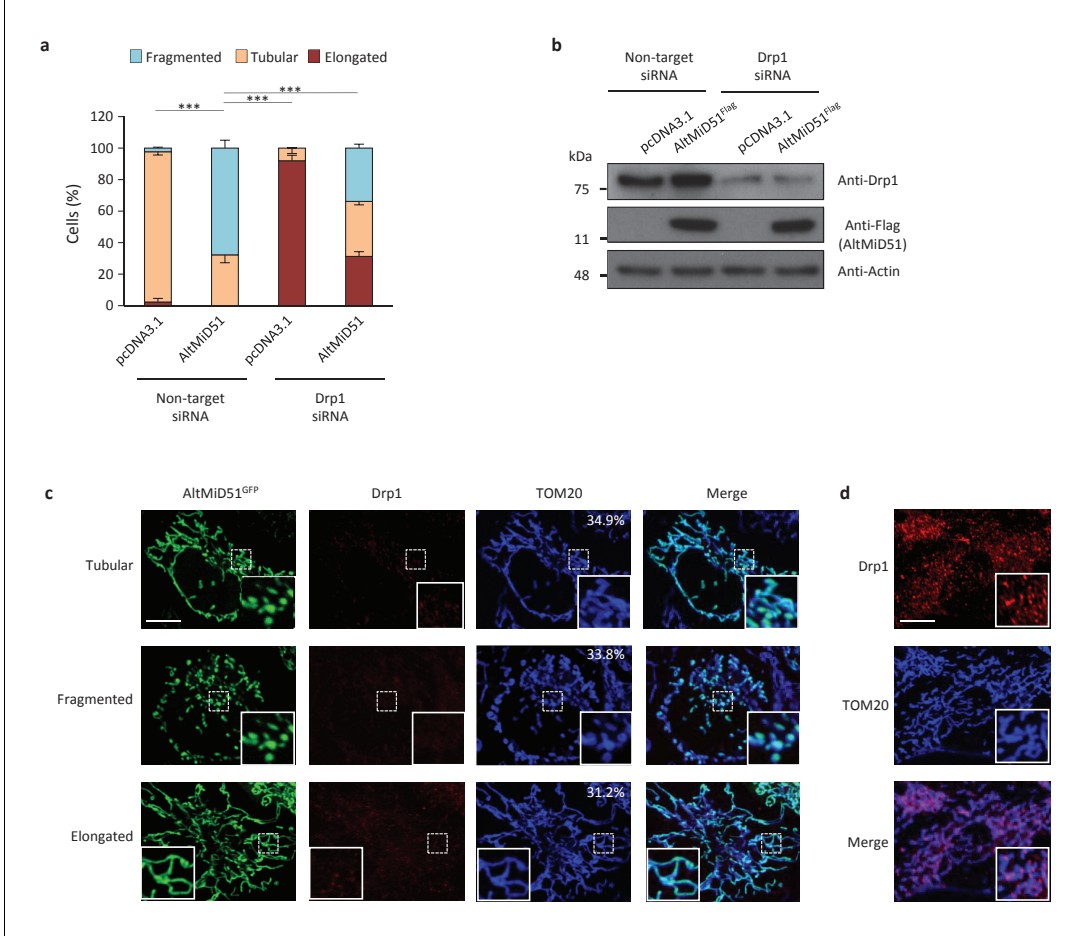

**Figure 13.** AltMiD51-induced mitochondrial fragmentation is dependent on Drp1. (**a**) Bar graphs show mitochondrial morphologies in HeLa cells treated with non-target or Drp1 siRNAs. Cells were mock-transfected (pcDNA3.1) or transfected with altMiD51[Flag]. Means of three independent experiments per condition are shown (100 cells for each independent experiment). ***p<0.0005 (Fisher's exact test) for the three morphologies between altMiD51 and the other experimental conditions. (**b**) HeLa cells treated with non-target or Drp1 siRNA were transfected with empty vector (pcDNA3.1) or altMiD51[Flag], as indicated. Proteins were extracted and analyzed by western blot with antibodies against the Flag tag (altMiD51), Drp1 or actin, as indicated. (**c**) Confocal microscopy of Drp1 knockdown cells transfected with altMiD51[GFP] immunostained with anti-TOM20 (blue channel) and anti-Drp1 (red channel) monoclonal antibodies. In each image, boxed areas are shown at higher magnification in the bottom right corner. % of cells with the indicated morphology is indicated on the TOM20 panels. Scale bar, 10 mm. (**d**) Control Drp1 immunostaining in HeLa cells treated with a non-target siRNA. For (**c**) and (**d**), laser parameters for Drp1 and TOM20 immunostaining were identical.

DOI: https://doi.org/10.7554/eLife.27860.047

The following source data is available for figure 13:

**Source data 1.** Mitochondrial morphologies in HeLa cells treated with non-target or Drp1 siRNAs.

DOI: https://doi.org/10.7554/eLife.27860.048

would follow the well-established models of gene evolution *de novo* (*Schlötterer, 2015*; *Knowles and McLysaght, 2009*; *Neme and Tautz, 2013*) in which new ORFs are transcribed and translated and have new functions or await the evolution of new functions by mutations. The numerous altORFs with no detectable protein domains may have originated this way from previously non-coding regions or in regions that completely overlap with CDS in other reading frames.

Detection is an important challenge in the study of small proteins. A TIS detected by ribosome profiling does not necessarily imply that the protein is expressed as a stable molecule, and proteomic analyses more readily detect large proteins that generate several peptides after enzymatic digestion. In addition, evolutionarily novel genes tend to be poorly expressed, again reducing the probability of detection (*Schlötterer, 2015*). Here, we used a combination of five search engines, thus increasing the confidence and sensitivity of hits compared to single-search-engine processing

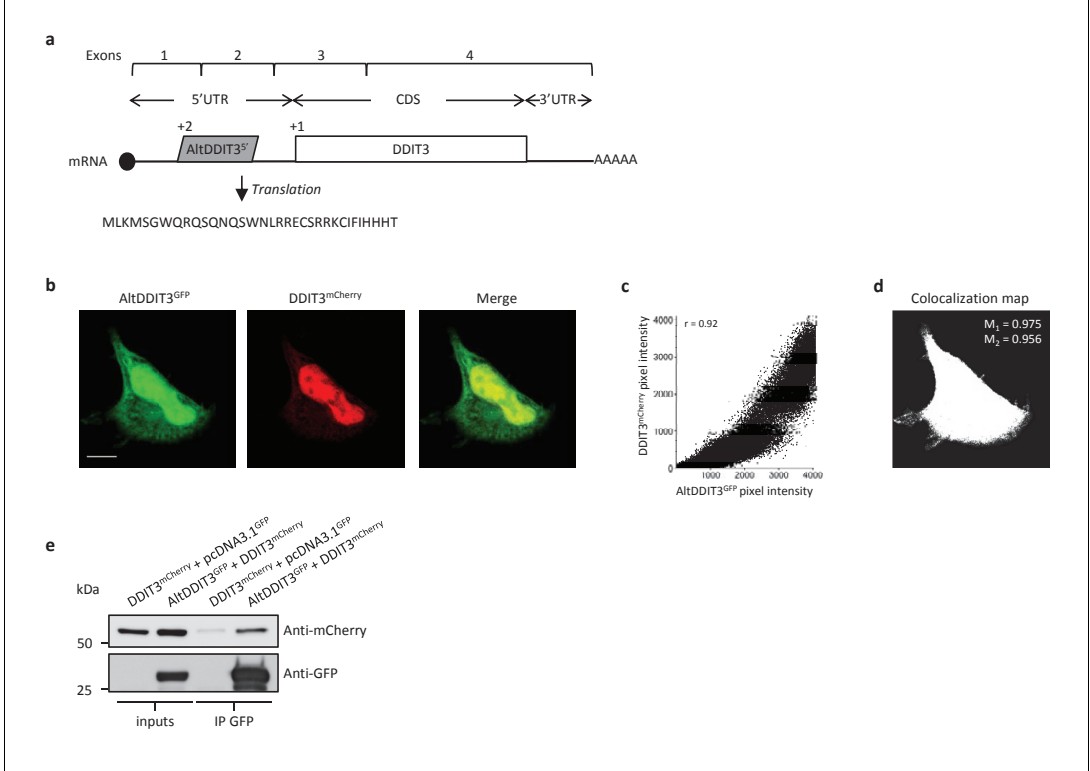

**Figure 14.** AltDDIT3$^{5'}$ co-localizes and interacts with DDIT3. (**a**) AltDDIT3$^{5'}$ coding sequence is located in exons 1 and 2 or the *DDIT3/CHOP/GADD153* gene and in the 5'UTR of the canonical mRNA (RefSeq NM_004083.5).+2 and+1 indicate reading frames. AltDDIT3 amino acid sequence is also shown. (**b**) Confocal microscopy analyses of HeLa cells co-transfected with altDDIT3$^{GFP}$ (green channel) and DDIT3$^{mCherry}$ (red channel). Scale bar, 10 μm. (**c, d**) Colocalization analysis of the images shown in (**b**) performed using the JACoP plugin (Just Another Co-localization Plugin) implemented in Image J software (two independent biological replicates). (**c**) Scatterplot representing 50% of green and red pixel intensities showing that altDDIT3$^{GFP}$ and DDIT3$^{mCherry}$ signal highly correlate (with Pearson correlation coefficient of 0.92 [p-value<0.0001]). (**d**) Binary version of the image shown in (**b**) after Costes' automatic threshold. White pixels represent colocalization events (p-value<0.001, based on 1000 rounds of Costes' randomization colocalization analysis). The associated Manders Correlation Coefficient, $M_1$ and $M_2$, are shown in the right upper corner. $M_1$ is the proportion of altDDIT3$^{GFP}$ signal overlapping DDIT3$^{mCherry}$ signal and $M_2$ is the proportion of DDIT3$^{mCherry}$ signal overlapping altDDIT3$^{GFP}$. (**e**) Representative immunoblot of co-immunoprecipitation with GFP-Trap agarose beads performed on HeLa lysates co-expressing DDIT3$^{mcherry}$ and altDDIT3$^{GFP}$ or DDIT3$^{mcherry}$ with pcDNA3.1$^{GFP}$ empty vector (two independent experiments).
DOI: https://doi.org/10.7554/eLife.27860.049

The following figure supplements are available for figure 14:

**Figure supplement 1.** Protein immunoblot showing the expression of different constructs in HeLa cells.
DOI: https://doi.org/10.7554/eLife.27860.050

**Figure supplement 2.** Colocalization of altDDIT3 with DDIT3.
DOI: https://doi.org/10.7554/eLife.27860.051

(*Vaudel et al., 2015*; *Shteynberg et al., 2011*). This strategy led to the detection of several thousand alternative proteins. However, ribosome profiling and MS have technical caveats and the comprehensive contribution of small proteins to the proteome will require more efforts, including the development of new tools such as specific antibodies.

Only a relatively small percentage of alternative proteins (22.6%) are functionally annotated with Interpro signatures, compared to reference proteins (96.9%). An obvious explanation is the small size of alternative proteins with a median size of 45 amino acids, which may not be able to accommodate large domains. It has been proposed that small proteins may be precursors of new proteins but require an elongation of their coding sequence before they display a useful cellular activity (*Carvunis et al., 2012*; *Couso and Patraquim, 2017*). According to this hypothesis, it is possible that protein domains appear only after elongation of the coding sequence. Alternatively, InterPro domains were identified by investigating the reference proteome, and alternative proteins may have

new domains and motifs that remain to be characterized. Finally, an unknown fraction of predicted altORFs may not be translated or may code for non-functional peptides.

In conclusion, our deep annotation of the transcriptome reveals that a large number of small eukaryotic proteins, possibly even the majority, are still not officially annotated. Our results with alt-MiD51, altDDIT3, and some zinc-finger proteins also suggest that some small and large proteins coded by the same mRNA may cooperate by regulating each other's function or by functioning in the same pathway, confirming the few examples in the literature of unrelated proteins encoded in the same genes and functionally cooperating (*Table 3*) (*Quelle et al., 1995*; *Abramowitz et al., 2004*; *Bergeron et al., 2013*; *Lee et al., 2014*; *Yosten et al., 2016*). To determine whether or not this functional cooperation is a general feature of small/large protein pairs encoded in the same gene will require more experimental evidence.

## Materials and methods

### Generation of alternative open-reading frames and alternative protein databases

Throughout this manuscript, annotated protein coding sequences and proteins in current databases are labeled annotated coding sequences or CDSs and reference proteins, respectively. For simplicity reasons, predicted alternative protein coding sequences are labeled alternative open-reading frames or altORFs.

To generate MySQL databases containing the sequences of all predicted alternative proteins translated from reference annotation of different organisms, a computational pipeline of Perl scripts was developed as previously described with some modifications (*Vanderperre et al., 2013*). Genome annotations for *H. sapiens* (release hg38, Assembly: GCF_000001405.26), *P. troglodytes* (Pan_troglodytes-2.1.4, Assembly: GCF_000001515.6), *M. musculus* (GRCm38.p2, Assembly: GCF_000001635.22), *D. melanogaster* (release 6, Assembly: GCA_000705575.1), *C. elegans* (WBcel235, Assembly: GCF_000002985.6) and *S. cerevisiae* (Sc_YJM993_v1, Assembly: GCA_000662435.1) were downloaded from the NCBI website (http://www.ncbi.nlm.nih.gov/genome). For *B. taurus* (release UMD 3.1.86), *X. tropicalis* (release JGI_4.2) and *D. rerio* (GRCz10.84), genome annotations were downloaded from Ensembl (http://www.ensembl.org/info/data/ftp/). Each annotated transcript was translated in silico with Transeq (*Rice et al., 2000*). All ORFs starting with an AUG and ending with a stop codon different from the CDS, with a minimum

**Table 3.** Examples of proteins encoded in the same gene and functionally interacting

| Gene | Polypeptides[*] | Reference | altORF localization | altORF size aa | Conservation | Summary of functional relationship with the annotated protein |
|------|------------|-----------|---------------------|----------------|--------------|----------------------------------------------------------------|
| CDKN2A, INK4 | Cyclin-dependent kinase inhibitor 2A or p16-INK4 (P42771), and p19ARF (Q8N726) | (61) | 5'UTR | 169 | Human, mouse | the unitary inheritance of p16INK4a and p19ARF may underlie their dual requirement in cell cycle control. |
| GNAS, XLalphas | Guanine nucleotide-binding protein G(s) subunit alpha isoforms XLαs (Q5JWF2) and Alex (P84996) | (62) | 5'UTR | +700 | Human, mouse, rat | Both subunits transduce receptor signals into stimulation of adenylyl cyclase. |
| ATXN1 | Ataxin-1 (P54253) and altAtaxin-1 | (63) | CDS | 185 | Human, chimpanzee, cow | Direct interaction |
| Adora2A | A2A adenosine receptor (P30543) and uORF5 | (64) | 5'UTR | 134 | Human, chimpanzee, rat, mouse | A2AR stimulation increases the level of the uORF5 protein via post-transcriptional regulation. |
| AGTR1 | Angiotensin type 1a receptor (P25095) and PEP7 | (65) | 5'UTR | 7 | Highly conserved across mammalian species | Inhibits non-G protein-coupled signalling of angiotensin II, without altering the classical G protein-coupled pathway activated by the ligand. |

[*]The UniProtKB accession is indicated when available.

DOI: https://doi.org/10.7554/eLife.27860.052

length of 30 codons (including the stop codon) and identified in a distinct reading frame compared to the annotated CDS when overlapping the CDS, were defined as altORFs.

An additional quality control step was performed to remove initially predicted altORFs with a high level of identity with reference proteins. Such altORFs typically start in a different coding frame than the reference protein but through alternative splicing, end with the same amino acid sequence as their associated reference protein. Using BLAST, altORFs overlapping CDSs chromosomal coordinates and showing more than 80% identity and overlap with an annotated CDS were rejected.

AltORF localization was assigned according to the position of the predicted translation initiation site (TIS): altORFs$^{5'}$, altORFs$^{CDS}$ and altORFs$^{3'}$ are altORFs with TISs located in 5'UTRs, CDSs and 3'UTRs, respectively. Non-coding RNAs (ncRNAs) have no annotated CDS and all ORFs located within ncRNAs are labeled altORFs$^{nc}$.

The presence of the simplified Kozak sequence (A/GNNATGG) known to be favorable for efficient translation initiation was also assessed for each predicted altORF (*Kozak, 2002*).

## Identification of TISs

The global aggregates of initiating ribosome profiles data were obtained from the initiating ribosome tracks in the GWIPS-viz genome browser (*Michel et al., 2014*) with ribosome profiling data collected from five large-scale studies (*Lee et al., 2012*; *Ji et al., 2015*; *Fritsch et al., 2012*; *Stern-Ginossar et al., 2012*; *Gao et al., 2015*). Sites were mapped to hg38 using a chain file from the UCSC genome browser (http://hgdownload.soe.ucsc.edu/goldenPath/hg19/liftOver/hg19ToHg38.over.chain.gz) and CrossMap v0.1.6 (RRID:SCR_001173). Similar to the methods used in these studies, an altORF is considered as having an active TIS if it is associated with at least ten reads at one of the seven nucleotide positions of the sequence NNNAUGN (AUG is the predicted altORF TIS). An additional recent study was also included in our analysis (*Raj et al., 2016*). In this study, a threshold of 5 reads was used. Raw sequencing data for ribosome protected fragments in harringtonine treated cells were aligned to the human genome (GRCh38) using bowtie2 (2.2.8) (*Langmead and Salzberg, 2012*). Similar to the method used in this work, altORFs with at least five reads overlapping one position in the kozak region were considered as having an experimentally validated TIS.

## Generation of shuffled transcriptomes

Each annotated transcript was shuffled using the Fisher-Yates shuffle algorithm. In CDS regions, all codons were shuffled except the initiation and stop codons. For mRNAs, we shuffled the 5'UTRs, CDSs and 3'UTRs independently to control for base composition. Non-coding regions were shuffled at the nucleotide level. The resulting shuffled transcriptome has the following features compared to hg38: same number of transcripts, same transcripts lengths, same nucleotide composition, and same amino-acid composition for the proteins translated from the CDSs. Shuffling was repeated 100 times and the results are presented with average values and standard deviations. The total number of altORFs is 539,134 for hg38, and an average of 489,073 for shuffled hg38. AltORFs and kozak motifs in the 100 shuffled transcriptomes were detected as described above for hg38.

## Identification of paralogs/orthologs in alternative proteomes

Both alternative and reference proteomes were investigated. Pairwise ortholog and paralog relationships between the human proteomes and the proteomes from other species, were calculated using an InParanoid-like approach (*Sonnhammer and Östlund, 2015*), as described below (RRID:SCR_006801). The following BLAST (RRID:SCR_001010) procedure was used. Comparisons using our datasets of altORFs/CDS protein sequences in multiple FASTA formats from *Saccharomyces cerevisiae, Caenorhabditis elegans, Drosophila melanogaster, Danio rerio, Xenopus tropicalis Bos taurus, Mus musculus, Pan troglodytes, Homo sapiens* were performed between each pair of species (*Homo sapiens* against the other species), involving four whole proteome runs per species pair: pairwise comparisons (organism A vs organism B, organism B vs organism A), plus two self-self runs (organism A vs organism A, organism B vs organism B). BLAST homology inference was accepted when the length of the aligned region between the query and the match sequence equalled or exceeded 50% of the length of the sequence, and when the bitscore reached a minimum of 40 (*Remm et al., 2001*). Orthologs were detected by finding the mutually best scoring pairwise hits (reciprocal best

hits) between datasets A-B and B-A. The self-self runs were used to identify paralogy relationships as described (*Sonnhammer and Östlund, 2015*).

## Analysis of third codon position (wobble) conservation

Basewise conservation scores for the alignment of 100 vertebrate genomes including *H. sapiens* were obtained from UCSC genome browser (http://hgdownload.soe.ucsc.edu/goldenPath/hg38/phyloP100way/) (RRID:SCR_012479). Conservation PhyloP scores relative to each nucleotide position within codons were extracted using a custom Perl script and the Bio-BigFile module version 1.07 (see code file). The PhyloP conservation score for the wobble nucleotide of each codon within the CDS was extracted. For the 53,862 altORFs completely nested inside 20,814 CDSs, the average PhyloP score for wobble nucleotides within the altORF region was compared to the average score for the complete CDS. To generate controls, random regions in CDSs with a similar length distribution as altORFs were selected and PhyloP scores for wobble nucleotides were extracted. We compared the differences between altORF and CDS PhyloP scores (altORF PhyloP – CDS PhyloP) to those generated based on random regions. We identified expected quantiles of the differences ('DQ' column in the table), and compared these to the observed differences. Because there was greater conservation of wobble nucleotide PhyloP scores within altORFs regions located farther from the center of their respective genes ($r$ = 0.08, p<0.0001), observed differences were adjusted using an 8 knot cubic basis spline of percent distance from center. These observed differences were also adjusted for site-specific signals as detected in the controls.

## Human alternative protein classification and in silico functional annotation

### Repeat and transposable element annotation

RepeatMasker, a popular software to scan DNA sequences for identifying and classifying repetitive elements (RRID:SCR_012954), was used to investigate the extent of altORFs derived from transposable elements (*Tarailo-Graovac and Chen, 2009*). Version 3-3-0 was run with default settings.

### Alternative protein analysis using InterProScan (RRID:SCR_005829)

InterProScan combines 15 different databases, most of which use Hidden Markov models for signature identification (*Jones et al., 2014*). Interpro merges the redundant predictions into a single entry and provides a common annotation. A recent local version of InterProScan 5.14–53.0 was run using default parameters to scan for known protein domains in alternative proteins. Gene ontology (GO) and pathway annotations were also reported if available with -goterm and -pa options. Only protein signatures with an E-value $<10^{-3}$ were considered.

We classified the reported InterPro hits as belonging to one or several of three clusters; (1) alternative proteins with InterPro entries; (2) alternative proteins with signal peptides (SP) and/or transmembrane domains (TM) predicted by at least two of the three SignalP, PHOBIUS, TMHMM tools and (3) alternative proteins with other signatures.

The GO terms assigned to alternative proteins with InterPro entries were grouped and categorized into 13 classes within the three ontologies (cellular component, biological process, molecular function) using the CateGOrizer tool (*Na et al., 2014*) (RRID:SCR_005737).

Each unique alternative protein with InterPro entries and its corresponding reference protein (encoded in the same transcript) were retrieved from our InterProscan output. Alternative and reference proteins without any InterPro entries were ignored. The overlap in InterPro entries between alternative and reference proteins was estimated as follows. We went through the list of alternative/reference protein pairs and counted the overlap in the number of entries between the alternative and reference proteins as 100*intersection/union. All reference proteins and the corresponding alternative proteins were combined together in each comparison so that all domains of all isoforms for a given reference protein were considered in each comparison. The random distribution of the number of alternative/reference protein pairs that have at least one identical InterPro entry was computed by shuffling the alternative/reference protein pairs and calculating how many have at least one identical InterPro entry. This procedure was repeated 1000 times. Finally, we compared the number and identity of co-ocurrence of InterPro entries in a two-dimensional matrix to illustrate which Interpro entries are identical in alternative/reference protein pairs. In many instances,

including for zinc-finger coding genes, InterPro entries in alternative/reference protein pairs tend to be related when they are not identical.

## Mass spectrometry identification

Wrapper Perl scripts were developed for the use of SearchGUI v2.0.11 (*Vaudel et al., 2011*) (RRID: SCR_012054) and PeptideShaker v1.1.0 (*Vaudel et al., 2015*) (RRID:SCR_002520) on the *Université de Sherbrooke's* 39,168 core high-performance *Mammouth Parallèle two* computing cluster (https:// www.computecanada.ca/research-portal/accessing-resources/available-resources/). SearchGUI was configured to run the following proteomics identification search engines: X!Tandem (*Craig and Beavis, 2004*), MS-GF+ (*Kim and Pevzner, 2014*), MyriMatch (*Tabb et al., 2007*), Comet (*Eng et al., 2013*), and OMSSA (*Geer et al., 2004*). SearchGUI parameters were set as follow: maximum precursor charge, 5; maximum number of PTM per peptide, 5; X!Tandem minimal fragment m/z, 140; removal of initiator methionine for Comet, 1. A full list of parameters used for SearchGUI and PeptideShaker is available in *Supplementary file 2*, sheet 1. For PXD000953 dataset (*Rosenberger et al., 2014*), precursor and fragment tolerance were set 0.006 Da and 0.1 Da respectively, with carbamidomethylation of C as a fixed modification and Nter-Acetylation and methionine oxidation as variable modifications. For PXD000788 (*Tong et al., 2014*) and PXD000612 (*Sharma et al., 2014*) datasets, precursor and fragment tolerance were set to 4.5 ppm and 0.1 Da, respectively, with carbamidomethylation of cysteine as a fixed modification and Nter-Acetylation, methionine oxidation and phosphorylation of serine, threonine and tyrosine as variable modifications. For PXD002815 dataset (*Hein et al., 2015*), precursor and fragment tolerance were set to 4.5 ppm and 0.1 Da, respectively, with carbamidomethylation of cysteine as a fixed modification and Nter-Acetylation and methionine oxidation as variable modifications.Datasets were searched using a target-decoy approach against a composite database composed of a target database [Uniprot canonical and isoform reference proteome (16 January 2015) for a total of 89,861 sequences + custom alternative proteome resulting from the in silico translation of all human altORFs (available to download at https://www.roucoulab.com/p/downloads)], and their reverse protein sequences from the target database used as decoys. In order to separate alternative and reference proteins for FDR analyses, PeptideShaker output files were extracted with target and decoy hits. PSMs matching reference target or decoy proteins were separated from those matching alternative targets or decoys as previously described (*Menschaert and Fenyö, 2017*; *Woo et al., 2014*). PSMs that matched both reference and alternative proteins were automatically moved to the reference database group. PSMs were then ranked according to their PeptideShaker score and filtered at 1% FDR separately. Validated PSMs were selected to group proteins using proteoQC R tool (*Gatto et al., 2015*), and proteins were separately filtered again using a 1% FDR cut-off.

Only alternative proteins identified with at least one unique and specific peptide were considered valid (*Vaudel et al., 2015*). Any peptide matching both a canonical (annotated in Uniprot) and an alternative protein was attributed to the canonical protein. For non-unique peptides, that is peptides matching more than one alternative protein, the different accession IDs are indicated in the MS files. For subsequent analyses (e.g. conservation, protein signature…), only one protein is numbered in the total count of alternative proteins; we arbitrarily selected the alternative protein with the lowest accession ID.

Peptides matching proteins in a protein sequence database for common contaminants were rejected (*Perkins et al., 1999*).

For spectral validation (*Figure 5—figure supplements 1*, *2*, *3* and *4*), synthetic peptides were purchased from the peptide synthesis service at the *Université de Sherbrooke*. Peptides were solubilized in 10% acetonitrile, 1% formic acid and directly injected into a Q-Exactive mass spectrometer (Thermo Scientific) via an electro spray ionization source (Thermo Scientific, Canada). Spectra were acquired using Xcalibur 2.2 (RRID:SCR_014593) at 70,000 resolution with an AGC target of 3e6 and HCD collision energy of 25. Peaks were assigned manually by comparing monoisotopic m/z theoretical fragments and experimental (PeptideShaker) spectra.

In order to test if the interaction between alternative zinc-finger/reference zinc-finger protein pairs (encoded in the same gene) may have occurred by chance only, all interactions between alternative proteins and reference proteins were randomized with an in-house randomization script. The number of interactions with reference proteins for each altProt was kept identical as the number of

observed interactions. The results indicate that interactions between alternative zinc-finger/reference zinc-finger protein pairs did not occur by chance ($p < 10^{-6}$) based on 1 million binomial simulations; highest observed random interactions between alternative zinc-finger proteins and their reference proteins = 3 (39 times out of 1 million simulations), compared to detected interactions = 5.

## Code availability
Computer codes are available upon request with no restrictions.

## Data availability
Alternative protein sequence databases for different species can be accessed at https://www.rou-coulab.com/p/downloads with no restrictions.

## Cloning and antibodies
Human Flag-tagged altMiD51(WT) and altMiD51(LYR→AAA), and HA-tagged DrP1(K38A) were cloned into pcDNA3.1 (Invitrogen) using a Gibson assembly kit (New England Biolabs, E26115). The cDNA corresponding to human MiD51/MIEF1/SMCR7L transcript variant 1 (NM_019008) was also cloned into pcDNA3.1 by Gibson assembly. In this construct, altMiD51 and MiD51 were tagged with Flag and HA tags, respectively. MiD51$^{GFP}$ and altMiD51$^{GFP}$ were also cloned into pcDNA3.1 by Gibson assembly. For MiD51$^{GFP}$, a LAP tag (*Hein et al., 2015*) was inserted between MiD51 and GFP. gBlocks were purchased from IDT. Human altDDIT3$^{mCherry}$ was cloned into pcDNA3.1 by Gibson assembly using coding sequence from transcript variant 1 (NM_004083.5) and mCherry coding sequence from pLenti-myc-GLUT4-mCherry (Addgene plasmid # 64049). Human DDIT3$^{GFP}$ was also cloned into pcDNA3.1 by Gibson assembly using CCDS8943 sequence.

For immunofluorescence, primary antibodies were diluted as follow: anti-Flag (Sigma, F1804) 1/1000, anti-TOM20 (Abcam, ab186734) 1/500. For western blots, primary antibodies were diluted as follow: anti-Flag (Sigma, F1804) 1/1000, anti-HA (BioLegend, 901515) 1/500, anti-actin (Sigma, A5441) 1/10000, anti-Drp1 (BD Transduction Laboratories, 611112) 1/500, anti-GFP (Santa Cruz Biotechnology, sc-9996) 1/10000, anti-mCherry (Abcam, ab125096) 1/2000.

## Cell culture, immunofluorescence, knockdown and western blots
HeLa cells (ATCC CRM-CCL-2, authenticated by STR profiling, RRID:CVCL_0030) cultures tested negative for mycoplasma contamination (ATCC 30–1012K), transfections, immunofluorescence, confocal analyses and western blots were carried out as previously described (*Vanderperre et al., 2011*). Mitochondrial morphology was analyzed as previously described (*Palmer et al., 2011*). A minimum of 100 cells were counted (n = 3 or 300 cells for each experimental condition). Three independent experiments were performed.

For Drp1 knockdown, 25,000 HeLa cells in 24-well plates were transfected with 25 nM Drp1 SMARTpool: siGENOME siRNA (Dharmacon, Canada, M-012092-01-0005) or ON-TARGET plus Non-targeting pool siRNAs (Dharmacon, D-001810-10-05) with DharmaFECT one transfection reagent (Dharmacon, T-2001–02) according to the manufacturer's protocol. After 24 hr, cells were transfected with pcDNA3.1 or altMiD51, incubated for 24 hr, and processed for immunofluorescence or western blot. Colocalization analyses were performed using the JACoP plugin (Just Another Colocalization Plugin) (*Bolte and Cordelières, 2006*) implemented in Image J software.

## Immunoprecipitations
Immunoprecipitations experiments were conducted using GFP-Trap (ChromoTek, Germany) protocol with minor modifications. Briefly, cells were lysed with co-ip lysis buffer (0.5 % NP40, Tris-HCl 50 mM pH 7.5, NaCl 150 mM and two EDTA-free Roche protease inhibitors per 50 mL of buffer). After 5 mins of lysis on ice, lysate was sonicated twice at 11% amplitude for 5 s with 3 min of cooling between sonication cycles. Lysate was centrifuged, supernatant was isolated and protein content was assessed using BCA assay (Pierce). GFP-Trap beads were conditioned with lysis buffer. 40 μL of beads were added to 2 mg of proteins at a final concentration of 1 mg/mL. After overnight immunoprecipitation, beads were centrifuged at 5000 rpm for 5 min and supernatant was discarded. Beads were then washed three times with wash buffer (0.5 % NP40, Tris-HCl 50 mM pH 7.5, NaCl 200 mM and two EDTA-free Roche protease inhibitors per 50 mL of buffer) and supernatants were discarded.

Immunoprecipitated proteins were eluted from beads by adding 40 µL of Laemmli buffer and boiling at 95°C for 15 min. Eluate was split in halfs which were loaded onto 10% SDS-PAGE gels to allow western blotting of GFP and mCherry tagged proteins. 40 µg of initial lysates were loaded into gels as inputs.

## Mitochondrial localization, parameters and ROS production.

Trypan blue quenching experiment was performed as previously described (*Vanderperre et al., 2016*).

A flux analyzer (XF96 Extracellular Flux Analyzer; Seahorse Bioscience, Agilent technologies, Santa Clara, CA) was used to determine the mitochondrial function in HeLa cells overexpressing alt-MiD51$^{Flag}$. Cells were plated in a XF96 plate (Seahorse Biosciences) at $1 \times 10^4$ cells per well in Dulbecco's modified Eagle's medium supplemented with 10% FBS with antibiotics. After 24 hr, cells were transfected for 24 hr with an empty vector (pcDNA3.1) or with the same vector expressing alt-MiD51$^{Flag}$ with GeneCellin tranfection reagent according to the manufacturer's instructions. Cells were equilibrated in XF assay media supplemented with 25 mM glucose and 1 mM pyruvate and were incubated at 37°C in a $CO_2$-free incubator for 1 hr. Baseline oxygen consumption rates (OCRs) of the cells were recorded with a mix/wait/measure times of 3/0/3 min respectively. Following these measurements, oligomycin (1 µM), FCCP (0.5 µM), and antimycin A/rotenone (1 µM) were sequentially injected, with oxygen consumption rate measurements recorded after each injection. Data were normalized to total protein in each well. For normalization, cells were lysed in the 96-well XF plates using 15 µl/well of RIPA lysis buffer (1% Triton X-100, 1% NaDeoxycholate, 0.1% SDS, 1 mM EDTA, 50 mM Tris-HCl pH 7.5). Protein concentration was measured using the BCA protein assay reagent (ThermoFisher Scientific, Canada).

Reactive oxygen species (ROS) levels were measured using Cellular ROS/Superoxide Detection Assay Kit (Abcam #139476, Cambridge, MA). HeLa cells were seeded onto 96-well black/clear bottom plates at a density of 6,000 cells per well with four replicates for each condition. After 24 hr, cells were transfected for 24 hr with an empty vector (pcDNA3.1) or with the same vector expressing altMiD51$^{Flag}$ with GeneCellin according to the manufacturer's instruction. Cells were untreated or incubated with the ROS inhibitor (N-acetyl-L-cysteine) at 10 mM for 1 hr. Following this, the cells were washed twice with the wash solution and then labeled for 1 hr with the Oxidative Stress Detection Reagent (green) diluted 1:1000 in the wash solution with or without the positive control ROS Inducer Pyocyanin at 100 µM. Fluorescence was monitored in real time. ROS accumulation rate was measured between 1 and 3 hr following induction. After the assay, total cellular protein content was measured using BCA protein assay reagent (Pierce, Waltham, MA) after lysis with RIPA buffer. Data were normalized for initial fluorescence and protein concentration.

ATP synthesis was measured as previously described (*Vives-Bauza et al., 2007*) in cells transfected for 24 hr with an empty vector (pcDNA3.1) or with the same vector expressing altMiD51$^{Flag}$.

## Acknowledgements

This research was supported by CIHR grants MOP-137056 and MOP-136962 to XR; MOP-299432 and MOP-324265 to CL; a *Université de Sherbrooke* institutional research grant made possible through a generous donation by Merck Sharp and Dohme to XR; a FRQNT team grant 2015-PR-181807 to CL and XR; Canada Research Chairs in Functional Proteomics and Discovery of New Proteins to XR, in Evolutionary Cell and Systems Biology to CL and in Computational and Biological Complexity to AO; AAC is supported by a CIHR New Investigator Salary Award; MSS is a recipient of a *Fonds de Recherche du Québec – Santé* Research Scholar Junior 2 Career Award; VD is supported in part by fellowships from *Région Nord-Pas de Calais* and PROTEO; AAC, DJH, MSS and XR are members of the *Fonds de Recherche du Québec Santé*-supported *Centre de Recherche du Centre Hospitalier Universitaire de Sherbrooke*. We thank the staff from the Centre for Computational Science at the *Université de Sherbrooke*, Compute Canada and Compute Québec for access to the Mammouth supercomputer.

## Additional information

### Funding

| Funder | Grant reference number | Author |
|---|---|---|
| Canadian Institutes of Health Research | MOP-137056 | Xavier Roucou |
| Canada Research Chairs | | Aida Ouangraoua<br>Christian R Landry<br>Xavier Roucou |
| Fonds de Recherche du Québec - Nature et Technologies | 2015-PR-181807 | Christian R Landry<br>Xavier Roucou |
| Merck Sharp and Dohme | | Xavier Roucou |
| Canadian Institutes of Health Research | MOP-136962 | Xavier Roucou |

The funders had no role in study design, data collection and interpretation, or the decision to submit the work for publication.

### Author contributions

Sondos Samandi, Formal analysis, Investigation, Methodology, Writing—original draft, Writing—review and editing; Annie V Roy, Formal analysis, Investigation, Visualization, Methodology; Vivian Delcourt, Data curation, Formal analysis, Investigation, Visualization, Methodology, Writing—original draft, Writing—review and editing; Jean-François Lucier, Conceptualization, Resources, Formal analysis, Investigation, Methodology; Jules Gagnon, Benoît Vanderperre, Julie Motard, Formal analysis, Investigation, Methodology; Maxime C Beaudoin, Marc-André Breton, Isabelle Gagnon-Arsenault, Formal analysis, Investigation; Jean-François Jacques, Formal analysis, Supervision, Investigation, Methodology; Mylène Brunelle, Formal analysis, Supervision, Methodology, Writing—original draft, Writing—review and editing; Isabelle Fournier, Supervision, Methodology; Aida Ouangraoua, Conceptualization, Methodology; Darel J Hunting, Conceptualization, Methodology, Writing—original draft, Writing—review and editing; Alan A Cohen, Conceptualization, Formal analysis, Investigation, Methodology, Writing—original draft, Writing—review and editing; Christian R Landry, Conceptualization, Formal analysis, Investigation, Visualization, Methodology, Writing—original draft, Writing—review and editing; Michelle S Scott, Conceptualization, Supervision, Investigation, Methodology, Writing—original draft, Writing—review and editing; Xavier Roucou, Conceptualization, Resources, Data curation, Formal analysis, Supervision, Funding acquisition, Investigation, Visualization, Methodology, Writing—original draft, Project administration, Writing—review and editing

### Author ORCIDs

Jean-François Jacques http://orcid.org/0000-0002-0465-0313
Isabelle Fournier https://orcid.org/0000-0003-1096-5044
Xavier Roucou http://orcid.org/0000-0001-9370-5584

### Decision letter and Author response

Decision letter https://doi.org/10.7554/eLife.27860.082
Author response https://doi.org/10.7554/eLife.27860.083

## Additional files

### Supplementary files

• Supplementary file 1. 12,616 alternative proteins and 26,531 reference proteins with translation initiation sites detected by ribosome profiling after re-analysis of large-scale studies. Sheet 1: general information. Sheet 2: list of alternative proteins; sheet 3: pie chart of corresponding altORFs localization. Sheet 4: Sheet 2: list of reference proteins
DOI: https://doi.org/10.7554/eLife.27860.053

• Supplementary file 2. 4,872 alternative proteins detected by mass spectrometry (MS) after re-analysis of large proteomic studies. Sheet 1: MS identification parameters; sheet 2: raw MS output; sheet 3: list of detected alternative proteins; sheet 4: pie chart of corresponding altORFs localization.
DOI: https://doi.org/10.7554/eLife.27860.054

• Supplementary file 3. List of phosphopeptides.
DOI: https://doi.org/10.7554/eLife.27860.055

• Supplementary file 4. Linker sequences separating adjacent zinc finger motifs.
DOI: https://doi.org/10.7554/eLife.27860.056

• Supplementary file 5. 100 alternative proteins with 25% to 100% identity and 10% to 100% overlap with their reference protein pairs. Sheet 1: BlastP output and protein domains.
DOI: https://doi.org/10.7554/eLife.27860.057

• Supplementary file 6. 383 alternative proteins detected by mass spectrometry in the interactome of 118 zinc finger proteins. Sheet 1: MS identification parameters; sheet 2: raw MS output; sheet 3: list of detected alternative proteins.
DOI: https://doi.org/10.7554/eLife.27860.058

• Supplementary file 7. High-confidence list of predicted functional alternative proteins based on conservation and expression analyses. Sheet 1: high-confidence list in mammals; sheet 2: high-confidence list in in vertebrates.
DOI: https://doi.org/10.7554/eLife.27860.059

• Source code 1. Extraction of PhyloP scores.
DOI: https://doi.org/10.7554/eLife.27860.060

• Transparent reporting form
DOI: https://doi.org/10.7554/eLife.27860.061

## Major datasets
The following datasets were generated:

| Author(s) | Year | Dataset title | Dataset URL | Database, license, and accessibility information |
|---|---|---|---|---|
| Samandi S, Roy A V, Delcourt V, Lucier J F, Gagnon J, et al | 2016 | Homo sapiens alternative protein predictions based on RefSeq GRCh38 (hg38) based on assembly GCF_000001405.26 | https://roucoulab.com/p/downloads | Freely accessible |
| Samandi S, Roy A V, Delcourt V, Lucier J F, Gagnon J, et al | 2016 | Pan troglodytes alternative protein predictions based on annotation version 2.4.1 | https://roucoulab.com/p/downloads | Freely accessible |
| Samandi S, Roy A V, Delcourt V, Lucier J F, Gagnon J, et al | 2016 | Mus musculus alternative protein predictions based on annotation version GRCm38 | https://roucoulab.com/p/downloads | Freely accessible |
| Samandi S, Roy A V, Delcourt V, Lucier J F, Gagnon J, et al | 2016 | Bos taurus alternative protein predictions based on annotation version UMD3.1.86 | https://roucoulab.com/p/downloads | Freely accessible |
| Samandi S, Roy A V, Delcourt V, Lucier J F, Gagnon J, et al | 2016 | Xenopus tropicalis alternative protein predictions based on Ensembl annotation JGI_4.2 | https://roucoulab.com/p/downloads | Freely accessible |
| Samandi S, Roy A V, Delcourt V, Lucier J F, Gagnon J, et al | 2016 | Danio rerio alternative protein predictions based on Ensembl annotation Zv9.79 | https://roucoulab.com/p/downloads | Freely accessible |
| Samandi S, Roy A V, Delcourt V, Lucier J F, Gagnon J, et al | 2016 | Drosophila melanogaster alternative protein predictions based on annotation version 6.07 | https://roucoulab.com/p/downloads | Freely accessible |
| Samandi S, Roy A V, Delcourt V, Lucier J F, Gagnon J, et al | 2016 | Caenorhabditis elegans alternative protein predictions based on annotation version WS251 | https://roucoulab.com/p/downloads | Freely accessible |
| Samandi S, Roy A V, Delcourt V, Lucier J F, Gagnon J, et al | 2016 | Saccharomyces cerevisiae S288c alternative protein predictions based on RefSeq annotation R64 assembly GCF_000146045.2 | https://roucoulab.com/p/downloads | Freely accessible |

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
