## [Decision Letter]

Thank you for submitting your article "Deep transcriptome annotation suggests that small and large proteins encoded in the same genes often cooperate" for consideration by *eLife*. Your article has been reviewed by three peer reviewers, and the evaluation has been overseen by a Reviewing Editor and Diethard Tautz as the Senior Editor. The following individuals involved in review of your submission have agreed to reveal their identity: Chen Xie (Reviewer #3).

The reviewers have discussed the reviews with one another and the Reviewing Editor has drafted this decision to help you prepare a revised submission.

Summary:

This paper describes an analysis of Eukaryotic genomes to identify alternative open reading frames (altORFs). The paper posits the existence of a large number of altORFs. Evidence that these are indeed protein coding includes there existence as orthologs across several species and the apparent constraint on their evolution between species. The authors also describe how the proteins encoded within the same gene may participate in similar functions.

This interesting and important study should be suitable for publication in *eLife* after addressing the major issues outlined below. Because this is the first systematical work about altORF, and the altORF list in the paper may play an important role in future research, the results have to be very solid and strict.

Essential revisions:

1) In the generation of the altORF database the authors remove all sequences that share more than 80% identity with the reference proteome. Nevertheless, there are still questions related to the similarity between altORF and reference proteins originating from the same genes. The authors should examine whether there is any association between the% overlap and the functional similarity (in terms of protein domains, localization, function etc.). Even if the overlap is lower than 80%, protein pair may still share multiple features.

2) Previous proteogenomic studies that find novel protein sequences based on genomic data (including point mutations) show lower score distribution of the new sequences, compared to the reference proteome (Menschaert and Fenyo, and related papers). Together with the larger size of the alternative sequences, analyses normally results in high false discovery rate of alternative sequences. The authors should present the score distributions and examine whether such different distribution occurs also in their altORF. Based on these, they may require separating between the FDR filtrations of these databases (as done with genomic data).

3) The overlap between ribo-seq and MS-based proteomics is surprisingly low. The authors explain that limitation of ribo-seq mapping probably underestimates the identified ORFs. It should be explained why the MS data does not cover larger proportions of the Ribo-seq IDs. Deeper analysis of this comparison should determine whether these missing ORFs are expected to be more lowly expressed, or are these specific to the examined systems in each discipline. Importantly, in case that the examined systems are markedly different, the comparison might be meaningless. The authors should try to analyze similar systems to enable meaningful assessment of these approaches.

4) The authors studied the conservation of each human altORF or annotated CDS by checking whether it existed in other species, and found many conserved altORFs. They should consider the possibility that even a neutral ORF may stay intact until a disabling mutation is fixated. A short ORF has much lower probability to be disabled than a long ORF and stays intact for much longer time, and most of altORFs are very short. It will be better if the authors use a statistical model to estimate the disabling probability of an ORF with a specific length in a specific time. Then they may estimate how many altORFs are really under functional constraint.

5) The authors used PhyloP values representing 100 vertebrates basewise conservation of the third positions of altORF overlapping and non-overlapping regions in annotated CDS to study the selection pattern of altORF CDS, but they should only use altORF-annotated CDS pairs conserved in vertebrates instead of all of them. In addition, they can also compare the PhyloP values of the first and second positions with the third positions of altORF5', altORF3', and altORFnc conserved in vertebrates to investigate their selection pattern.

6) In their analyses of ribosome profiling data, the authors believed that a TIS with 10 or 5 reads is active. It may be an artifact because there are always reads in the 5' UTR of mRNA even there is no upstream ORF; and reads supporting the TIS of an altORFCDS may be just due to the translation of annotated CDS. That may explain why much more altORF5' and altORFCDS are supported than altORF3' and altORFnc. A more strict way is to check the periodicity of the reads, and there are already tools, such as RiboTaper or RibORF. The authors should use at least one of the tools to analyze and check whether they get the same pattern. In addition, the authors should also analyze the altORFs supported by proteomics data in the same way as Figure 5.

7) In their analyses of proteomics data, is it possible that the peptides of altORFs may also come from the microbe in human samples, such as parasites or infectious microbe? For example, randomly picking up a peptide "MKHIPSR" (Supplementary file 2, table "RAW MS OUTPUT", row 9), and running NCBI BLASTP against the nr database, found four perfect matches, including one from a protein in *Pseudomonas*, which is an infectious bacteria. This phenomena may be highly likely for short peptides. Unless this scenario is highly improbable, the authors could filter out the peptides, which can also be from the microbe in human in order to make the evidence more reliable.

8) Subsection "Evidence of functional coupling between reference and alternative proteins coded by the same genes", the authors claimed that "If there is functional cooperation or shared function, one would expect orthologous alternative-reference protein pairs to be co-conserved". However, functional cooperated proteins are not necessarily co-conserved. For example, a newly originated altORF can also interact with its annotated CDS; many co-conserved proteins do not interact with each other, e.g., there are tens of thousands of proteins co-conserved in human and chimpanzee and most of them do not cooperate. The co-conservation analysis may be informative for multiple gene birth and death events in a large phylogenic tree of dozens or hundreds of species, but not in this case.

9) Paragraph two of the same subsection, the authors claimed "Another mechanism that could functionally associate alternative and reference proteins from the same transcripts would be that they share protein domains". Proteins that share similar domains do not necessarily cooperate, and in contrast many proteins that do not share similar domains do interact nevertheless. The results about the zinc finger proteins and the two altORFs that are experimentally studied are interesting. However, it is better to remove the part about relation between co-conservation / domain sharing and functional cooperation.

10) Discussion first paragraph, the authors claimed that "underrepresentation of altORFs in repeat sequences" supported the functional role of altORFs. However, their altORFs were predicted from the transcriptional annotation that was probably underrepresented in repeat regions, and this observation may be simply caused by the biased locations of transcripts. They should also perform the same analysis of repeat regions with annotated CDS, and compare the results with those of altORFs. Combining this with the above two points, the statement of three lines of evidence: (6), (7), (8) should be removed.

11) Finally, the data at "https://roucoulab.com/p/downloads" only include simple information of altORFs. Considering that this altORF list may be important for further studies, as much information as possible should be provided in order to let others follow the work easier. We encourage the authors to include freely accessible and well-organized tables containing information about the 183,191 human altORFs and 51,818 annotated CDSs at their website. For each ORF/CDS, the authors should provide: ORF accession, ORF type (5', CDS, 3', nc), chromosome, start position, end position, sequence, transcript accession, conservation (chimpanzee, mouse,.[...], yeast), ribosome profiling evidence (each data set separately), proteomics evidence (each data set separately), predicted functional signatures. Perhaps more data? It should not require too much effort since the data are already stored in MySQL database.

[Editors' note: further revisions were requested prior to acceptance, as described below.]

Thank you for resubmitting your work entitled "Deep transcriptome annotation suggests that small and large proteins encoded in the same genes often cooperate" for further consideration at *eLife*. Your revised article has been favorably evaluated by Diethard Tautz (Senior editor), a Reviewing editor, and three reviewers.

Overall, the authors dealt very well with the points that have been raised about the original manuscript. The few that remain (detailed below) can be addressed without a second round of external review.

Reviewer #2:

Overall the authors properly addressed all my comments and edited the manuscript accordingly. In some cases they decided not to include the analyses in the revised manuscript.

Comment #1: Regarding the protein sequence overlap and association with function, I suggest to add a short description of the Results section (1-2 sentences) and the figure in the supplementary material.

Comment #2: The authors properly addressed the issue and edited the manuscript accordingly.

Comment #3: I think the authors did not fully understand the question. I referred to predicted proteins based on ribo-seq which are not identified by proteomics (and not vice versa). I accept their claim that this is not a major aim in the paper, but it could be still interesting to discuss the underlying causes for discrepancies. Nevertheless, I don't think it is absolutely required for the paper.

Minor comment #8: Since the quantitative aspect of the result is indeed not critical here, I accept their compromise which results from the availability of appropriate analytical software.

Altogether, in terms of the proteomic aspects, I find the manuscript improved compared to the previous version (mainly due to the FDR correction), and the rest has been addressed in the authors' reply.

Reviewer #3:

The authors addressed most of the comments well, besides they showed important results, including a large number of altORFs existing in the genome, and many of them probably encoding functional proteins due to that they were under functional constraint, or were expressed, or had functional domains, therefore, I think this paper could be accepted if the authors solve all remaining comments as follows well.

As I mentioned in essential revision point 8 and 9, which the authors also agreed, co-conservation or domain sharing does not mean functional interaction/cooperation. The co-conservation or domain sharing evidence in this paper was very weak and did not support the conclusion, functional cooperation. In the other way, even the few cases mentioned in the paper, which altORFs and annotated ORFs did functionally interact with each other do not show co-conservation or domain sharing. Simply changing "functional cooperation" to "functional relationship" makes no difference. The relevant parts in the manuscript should be removed, and the Title and Abstract should also be edited accordingly. Actually, the rest of the work is already very significant and solid.

---

## [Author Response]

Essential revisions:1) In the generation of the altORF database the authors remove all sequences that share more than 80% identity with the reference proteome. Nevertheless, there are still questions related to the similarity between altORF and reference proteins originating from the same genes. The authors should examine whether there is any association between the% overlap and the functional similarity (in terms of protein domains, localization, function etc.). Even if the overlap is lower than 80%, protein pair may still share multiple features.

In order to examine whether there is any association between the% overlap or identity and functional similarity between alternative-reference protein pairs (encoded in the same gene), we have blasted the 183,191 predicted alternative proteins against 54,498 reference proteins using BlastP. All altORFs with more than 80% identity and overlap had already been removed to generate our database (as indicated in the Materials and methods). We have found 100 (0.055%) alternative proteins with 25 to 100% identity and 10 to 100% overlap with their reference protein pairs. Among them, 20 (0.00054%) alternative proteins share the same InterPro signature with their respective reference proteins. The distribution of the percentage of sequence identity and overlap between alternative-reference protein pairs with (w/, n=20) or without (w/o, n=80) identical Interpro signature is shown in Figure 11—figure supplement 3. We observed no significant differences between both groups (p-value=0.6272; Kolmogorov Smirnov test). We conclude that there is no significant association between identity/overlap and functional relationship.

2) Previous proteogenomic studies that find novel protein sequences based on genomic data (including point mutations) show lower score distribution of the new sequences, compared to the reference proteome (Menschaert and Fenyo, and related papers). Together with the larger size of the alternative sequences, analyses normally results in high false discovery rate of alternative sequences. The authors should present the score distributions and examine whether such different distribution occurs also in their altORF. Based on these, they may require separating between the FDR filtrations of these databases (as done with genomic data).

As suggested by the reviewer, we compared the score distributions between reference proteins PSMs and previous combined FDR alternative proteins PSMs. We observed a significant difference in the distribution, even when comparing alternative and reference proteins with comparable numbers of peptides and PSM counts. Even though part of this difference is attributable to the small size of alternative proteins and thus reduced chances to produce high quality fragmentation products, we proceeded to separate alternative and reference proteins for FDR filtrations as previously described (PMID: 26670565, 25263569). This resulted in a reduced number of unique alternative protein hits of 4,872 (previously 10,362) and a total number of PSM of 15,491 (previously 40,478). The text, tables, figures and methods have been modified accordingly.

3) The overlap between ribo-seq and MS-based proteomics is surprisingly low. The authors explain that limitation of ribo-seq mapping probably underestimates the identified ORFs. It should be explained why the MS data does not cover larger proportions of the Ribo-seq IDs. Deeper analysis of this comparison should determine whether these missing ORFs are expected to be more lowly expressed, or are these specific to the examined systems in each discipline. Importantly, in case that the examined systems are markedly different, the comparison might be meaningless. The authors should try to analyze similar systems to enable meaningful assessment of these approaches.

We agree with the reviewer that the overlap is low and two reasons may explain this observation. (1) For the prediction of altORFs, the predicted TIS is the AUG that defines the longest altORF. It is possible that experimental TISs correspond to downstream in-frame AUGs. Alternatively, our TIS predictions are based on the presence of AUG codons only. Yet, a fraction of experimental TISs occur on non-AUG codons. In both cases, an erroneous assumption (wrong AUG or non-AUG start site) in the prediction necessarily results in the lack of detection of the altORF by ribosome profiling. (2) About 40% of predicted and detected altORFs are localized in 3'UTRs. Even though a technical issue preventing ribosome profiling in 3'UTRs was recently resolved, as indicated in the manuscript, but no experimental data for initiating ribosomes has been published yet with this 3'UTR-optimized protocol.

Our goal was to use a novel protein sequence database with unannotated (or alternative) ORFs (mostly but not exclusively small ORFs), and reanalyze independent public large-scale MS and ribo-seq data, to test if novel proteins expressed from these alternative ORFs can be detected. Our aim was not to compare two independent methods, ribo-seq and MS, to identify these novel proteins. Thus, we believe that our study provides a proof of principle that many unassigned peptides in MS data can be assigned to alternative proteins, and TISs identified in ribo-seq data can also match alternative ORFs. In this manuscript, we further focused on the functional annotation and the functional relationships between proteins encoded in the same genes. Nevertheless, it will be interesting in further studies to run both MS and ribo-seq experiments on similar systems (same cells, same tissues) in order to show the expression of these novel proteins with two independent methods.

4) The authors studied the conservation of each human altORF or annotated CDS by checking whether it existed in other species, and found many conserved altORFs. They should consider the possibility that even a neutral ORF may stay intact until a disabling mutation is fixated. A short ORF has much lower probability to be disabled than a long ORF and stays intact for much longer time, and most of altORFs are very short. It will be better if the authors use a statistical model to estimate the disabling probability of an ORF with a specific length in a specific time. Then they may estimate how many altORFs are really under functional constraint.

This is an interesting and important point, and our team exerted considerable effort discussing how we might successfully implement it. There are a number of operational challenges to designing a statistical model to estimate the probability that an ORF becomes disabled by a mutation. To start with, each ORF may have its own functionality and be subject to types of disabling mutations that cannot be easily identified by broad-scale bioinformatics approaches; this is likely to be particularly true for altORFs, which may represent understudied or novel protein classes. We thus started with the assumption that our model would look only at the introduction of a stop codon into a previously functioning sequence.

A proper model for this would minimally need to incorporate (a) the fact that different codons may appear with different frequencies in relation to nucleotide frequencies; (b) varying mutation rates depending on the nucleotide, codon, and larger genomic context; (c) different probabilities of arriving at a stop codon depending on the precursor codon; (d) varying mutation rates across the tree of life; (e) the probability that the stop codon is itself subsequently lost, so as to re-generate the detection of conservation in our analyses; and (f) the phylogenetic structure of the dataset. While it is certainly possible to construct such a model, it is also a formidable task we feel goes well beyond what is normally requested in the context of revising an article: it would require more effort than many entire publications, and could certainly not be achieved within the 2-month revision timeframe. For the moment, we have only generated the conservation data for 8 species (relative to humans), and this was itself a substantial task, but we would need many more and better phylogenetic resolution to run a convincing model.

We thus do not feel it is possible to run anything like a convincing model that would capture the probability that altORFs are disabled as a function of their length over evolutionary time, within the context of this article. Simplifying assumptions might be made, but they are likely to cause us to have substantial doubt as to the conclusions of the model. We have thus arrived at an intermediate solution we hope will prove satisfactory: we have examined the relationship between the length of alternative and reference proteins and their conservation in our existing data set. We do not think this analysis necessarily belongs in the manuscript itself (though we would be happy to include it in the Supplement if requested), but we do feel it should assuage any concerns that might be related to this point regarding the validity of our analyses, and we present the results here for the reviewers/editors.

Briefly, we plotted the number of species in which a protein (alternative or reference) was found conserved relative to humans (maximum of 8 for our eight species: yeast, nematodes, fruit flies, zebrafish, *Xenopus*, mice, cow, chimpanzee) against the length of the generated protein in codons. We did this for the full set of human reference proteins, the full set of alternative proteins, and several alternative protein subsets chosen to be progressively stronger candidates to code functional proteins. These additional categories were based on six criteria: (1) presence of a kozak sequence; (2) presence of a transmembrane sequence; (3) presence of a signaling peptide sequence; (4) presence of an interpro scan entry; (5) detection by mass spec; and (6) detection by TIS. The least stringent group was selected based on the presence of at least two of these criteria ("weak candidates", 6085 alternative proteins). The next most stringent was to require both TIS and mass spec detection ("TIS + MS", 467 alternative proteins). Third, we required 3 or more criteria ("Strong candidates", 278 alternative proteins). Fourth, we required TIS and mass spec and one additional criteria ("TIS + MS + 1"; 99 alternative proteins).

We also analyzed the relationship statistically. Linear models did not appear to fully capture the relationship, so we tried a quadratic model, which had a better fit to the data in all cases. We additionally tried a cubic spline to assess whether the quadratic model was sufficient; spline coefficient p-values and visual inspection of splines indicated that there was little improvement over the quadratic model, so we limited ourselves to the quadratic model.

We present only the visual results, as there is little additional information to be gained from the statistical details. Please keep in mind the varying samples sizes - particularly, the very large number of reference proteins and alternative proteins relative to the numbers of more restrictive alternative proteins categories. Author response image 1 shows that, as is known, CDSs are much longer than altORFs, and many more of them are fully conserved across all 8 species. However, the relationship between length and conservation in CDSs is strongly negatively quadratic, with highest conservation at intermediate lengths and a peak at about 350 codons. This same negative quadratic pattern was also apparent in all altORF categories, but was much less pronounced for the full altORF category than for the increasingly restrictive subsets. Notice how the peak of the quadratic curve becomes much sharper in increasingly stringent subsets.

We believe the quadratic relationship indicates substantial complexity in the relationship between length and conservation. On the one hand, the comment of the reviewer is certainly right that shorter sequences are less likely to disappear randomly, and on the other it is known that longer sequences tend to be better conserved for other reasons (PMID 12410938, 24379829). There are probably other factors at play as well that combine to produce the observed quadratic relationships. We believe the progressively more quadratic relationships for the progressively more stringent altORF categories indicate an increasing detection of functional proteins in these categories. In the full altORF category, a mix of non-functional sequences likely dilutes the signal, and produces a straighter, less quadratic relationship that is closer to the linear decline predicted by the comment. In the more stringent categories, the increasing conservation at intermediate lengths suggests that those categories are successfully identifying sequences that respond to the full range of underlying forces that produce the quadratic relationship in the CDSs.

Crucially, we note that the strong quadratic relationships imply that a model that controls for the appearance of a stop codon would be inappropriate in this context. There are other factors at work that intersect with length, and if we were to model the impact of length on the persistence of a sequence based solely on the risk of a stop codon without also modelling these other factors, we would certainly arrive at false or misleading results.

Lastly, we note that the possibility of a large number of false positives among the 183K alternative proteins presents challenges for detecting a generalized signal, and that any future models along these lines will likely have to use our approach of selecting candidate altORFs based on additional criteria.

5) The authors used PhyloP values representing 100 vertebrates basewise conservation of the third positions of altORF overlapping and non-overlapping regions in annotated CDS to study the selection pattern of altORF CDS, but they should only use altORF-annotated CDS pairs conserved in vertebrates instead of all of them. In addition, they can also compare the PhyloP values of the first and second positions with the third positions of altORF5', altORF3', and altORFnc conserved in vertebrates to investigate their selection pattern.

We would like first to highlight that this analysis was performed with altORFs completely nested within CDSs, as indicated in the manuscript. As requested by the reviewer, we have repeated this analysis with altORF-CDS pairs conserved in vertebrates. Since the reviewer mentions conserved altORFs-CDS pairs, we assumed he/she meant co-conserved pairs. In order to do that, we had to move up the co-conservation analysis from previous Figure 11 (section "Evidence of functional coupling between reference and alternative proteins coded by the same genes") to the "Conservation analyses" subsection (new Figure 3). The results of the new analysis appear in the new Figure 4, and the conclusions have not changed. We have modified the text and the figure legend accordingly.

We have moved the original Figure 3 to Figure 4—figure supplement 1.

With regards to the PhyloP values of the first and second positions with the third positions of altORF5', altORF3', and altORFnc conserved in vertebrates, we identified 30 co-conserved altORF5' and altORF3'. Since an altORFnc cannot be paired with a CDS (i.e. by definition, ncRNAs do not have an annotated CDS), altORFnc could not be included in this analysis. PhyloP values were extracted and plotted in Author response image 2 (graph A). We looked at conservation level in UTR regions outside co-conserved altORF5'-CDS and altORF3'-CDS pairs by examining phyloP values in 5 randomly picked non-altORF. Thus, for each altORF, we generated 5 random altORFs of the same length in the same UTR. These random UTRs do not overlap the non-random altORF. In most cases, average phyloP values were lower in the random altORFs compared to the non-random altORFs (graph B).

Among the 30 co-conserved altORF5' and altORF3', 6 altORFs (i.e. 20%) highlighted in red in the graphs display a clear pattern with PhyloP values for nucleotides 1 and 2 clearly above PhyloP values for nucleotide 3 (graph A), whilst this pattern completely disappears in the corresponding random altORFs (graph B).

Due to the low numbers of co-conserved altORF5'- and altORF3'-CDS pairs, we believe that it would be too speculative to draw a definitive conclusion and we have not included the results of this analysis in the revised manuscript.

**Author response image 2. respfig2:** 

6) In their analyses of ribosome profiling data, the authors believed that a TIS with 10 or 5 reads is active. It may be an artifact because there are always reads in the 5' UTR of mRNA even there is no upstream ORF; and reads supporting the TIS of an altORFCDS may be just due to the translation of annotated CDS. That may explain why much more altORF5' and altORFCDS are supported than altORF3' and altORFnc. A more strict way is to check the periodicity of the reads, and there are already tools, such as RiboTaper or RibORF. The authors should use at least one of the tools to analyze and check whether they get the same pattern. In addition, the authors should also analyze the altORFs supported by proteomics data in the same way as Figure 5.

We would like to mention that we did not select the 10 and 5 reads thresholds. These thresholds were those selected by the authors of the manuscripts used in our analyses. Since these manuscripts have already been peer-reviewed, we believe that it is reasonable to use what they believed are valid TISs, i.e. TISs with a minimum of 5 or 10 reads, depending on the study. Our analyses consisted in identifying TISs experimentally detected in those studies, which match predicted altORFs start sites.

With regards to the characteristic three-nucleotide periodicity of the reads, this applies only to elongating ribosomes, but not to initiating ribosomes (RiboTaper: PMID 26657557; RibORF: PMID 26687005. For a review: PMID 27015305). We analyzed ribosome profiling data for initiating ribosomes only. We did not analyze ribosome profiling data for elongating ribosome.

We have analyzed the altORFs supported by proteomics data in the same way as Figure 5, as requested by the reviewer (Figure 6). We have added the following text in the results: The majority of these proteins are coded by altORF^CDS^, but there are also significant contributions of altORF^3'^, altORF^nc^ and altORF^5'^ (Figure 6).

7) In their analyses of proteomics data, is it possible that the peptides of altORFs may also come from the microbe in human samples, such as parasites or infectious microbe? For example, randomly picking up a peptide "MKHIPSR" (Supplementary file 2, table "RAW MS OUTPUT", row 9), and running NCBI BLASTP against the nr database, found four perfect matches, including one from a protein in Pseudomonas, which is an infectious bacteria. This phenomena may be highly likely for short peptides. Unless this scenario is highly improbable, the authors could filter out the peptides, which can also be from the microbe in human in order to make the evidence more reliable.

We have blasted all detected peptides against the proteome from several organisms.

*-Pseudomonas*:

11,811 sequences from Uniprot (http://www.uniprot.org/uniprot/?query=*Pseudomonas*%20AND%20organism:%22pseudomonas%22&fil=reviewed:yes&sort=score)

994100 sequences from NCBI (https://www.ncbi.nlm.nih.gov/protein/?term=*Pseudomonas*)

-Deinococcus: 202,035 sequences from NCBI (https://www.ncbi.nlm.nih.gov/protein/?term=deinococcus)

-*Escherichia coli*:

4,446 sequences from Uniprot (http://www.uniprot.org/uniprot/?query=escherichia%20coli%20organism:%22escherichia%20coli%22&fil=organism%3A%22Escherichia+coli+%28strain+K12%29+%5B83333%5D%22+AND+reviewed%3Ayes&sort=score)

3262888 sequences from NCBI (https://www.ncbi.nlm.nih.gov/protein/?term=Escherichia+Coli)

Archaeon: 574084 sequences from NCBI (https://www.ncbi.nlm.nih.gov/protein/?term=Archaeon)

Viruses: 17,374 sequences (173 different species) from Uniprot (http://www.uniprot.org/uniprot/?query=viruses&fil=reviewed%3Ayes&sort=score)

Importantly, these blast outputs did not return any hits with 100 percent identity and 100% overlap. Hence, the detected peptides in our analyses are not present in the NCBI and Uniprot fasta files containing the up-to-date proteome of these organisms.

When performing an online blastp against the nr database as the reviewer did, 100 percent identity and overlap peptides may be identified (e.g. peptide MKHIPSR); however, they match hypothetical proteins that do not exist in the NCBI and Uniprot proteome ftp files. In other words, these hypothetical proteins are not annotated and are not currently considered in the annotated proteomes. Some of them may be annotated in the future or may never be annotated.

8) Subsection "Evidence of functional coupling between reference and alternative proteins coded by the same genes", the authors claimed that "If there is functional cooperation or shared function, one would expect orthologous alternative-reference protein pairs to be co-conserved". However, functional cooperated proteins are not necessarily co-conserved. For example, a newly originated altORF can also interact with its annotated CDS; many co-conserved proteins do not interact with each other, e.g., there are tens of thousands of proteins co-conserved in human and chimpanzee and most of them do not cooperate. The co-conservation analysis may be informative for multiple gene birth and death events in a large phylogenic tree of dozens or hundreds of species, but not in this case.

We agree with the reviewer that co-conservation does not necessarily mean interaction or cooperation. Nevertheless, the assumption that proteins that function together in a pathway or in a complex are likely to be co-conserved more often than expected inspired the development of different computational methods to infer protein cooperation and build interaction networks (PMID 20334628, 17803817, 15128449). What we actually meant is that co-conservation more often than expected of a small and a large protein coded in the same gene would not necessarily mean functional cooperation or interaction, but would nevertheless support a possible relationship. In the revised manuscript, we use the term *functional relationship* rather than functional cooperation/interaction. We have also added the following sentence: "The functional associations discussed here are potential functional interactions that do not necessarily imply physical interactions."

9) Paragraph two of the same subsection, the authors claimed "Another mechanism that could functionally associate alternative and reference proteins from the same transcripts would be that they share protein domains". Proteins that share similar domains do not necessarily cooperate, and in contrast many proteins that do not share similar domains do interact nevertheless. The results about the zinc finger proteins and the two altORFs that are experimentally studied are interesting. However, it is better to remove the part about relation between co-conservation / domain sharing and functional cooperation.

We agree with this comment that functional domain sharing does not imply cooperation or interaction. The functional associations discussed here are potential functional interactions, not direct physical interactions. Indeed, domains determine the function of proteins, and domain annotation is actually used by function prediction programs (PMID 16772267). In the absence of experimental evidence, inference of shared function can be obtained by comparative analyses of domains and motifs (PMID 27899635, 18177498, 15980588). Also, two proteins sharing common domains or belonging to the same family are more likely to be functionally linked, specifically have related functions with reference to molecular function and biological process (e.g. PMID 16632496).

We have changed the wording in order to avoid any misinterpretation. For example, we have systematically changed functional cooperation to functional relationship in the revised manuscript. In addition, we have added the following sentence earlier in the text (see response to comment 8): "The functional associations discussed here are potential functional interactions that do not necessarily imply physical interactions."

10) Discussion first paragraph, the authors claimed that "underrepresentation of altORFs in repeat sequences" supported the functional role of altORFs. However, their altORFs were predicted from the transcriptional annotation that was probably underrepresented in repeat regions, and this observation may be simply caused by the biased locations of transcripts. They should also perform the same analysis of repeat regions with annotated CDS, and compare the results with those of altORFs. Combining this with the above two points, the statement of three lines of evidence: (6), (7), (8) should be removed.

We did not find evidence in the literature that recent transcriptional annotations are probably underrepresented in repeat regions. In contrast, a large proportion of the transcripts in the human genome are thought to be initiated from repetitive elements. The FANTOM4 project reported that 31.4% of cap-selected human transcripts initiate within repetitive elements (PMID 19377475). Another study on the landscape of transcription in human cells reported a figure of 18% (PMID 22955620). Other studies show that repeat regions are actively transcribed (e.g. PMID 25012247).

We have performed the analysis of repeat regions with CDSs. In agreement with the reviewer\x92s comment, the results shown in the revised Figure 1—figure supplement 1 indicate that there are 4 times more altORFs that CDSs in repeat regions (9.83% vs 2.45%). We have therefore removed this argument from the statements in the Discussion.

11) Finally, the data at "https://roucoulab.com/p/downloads" only include simple information of altORFs. Considering that this altORF list may be important for further studies, as much information as possible should be provided in order to let others follow the work easier. We encourage the authors to include freely accessible and well-organized tables containing information about the 183,191 human altORFs and 51,818 annotated CDSs at their website. For each ORF/CDS, the authors should provide: ORF accession, ORF type (5', CDS, 3', nc), chromosome, start position, end position, sequence, transcript accession, conservation (chimpanzee, mouse, [...], yeast), ribosome profiling evidence (each data set separately), proteomics evidence (each data set separately), predicted functional signatures. Perhaps more data? It should not require too much effort since the data are already stored in MySQL database.

We agree with the referee. We have added an xls file on the web site and we have added the following sentence at the start of the Discussion: "We have provided the first functional annotation of altORFs with a minimum size of 30 codons in different genomes. The comprehensive annotation of *H sapiens* altORFs is freely available to download at https://www.roucoulab.com/p/downloads (H sapiens alternative proteins_functional annotation)."

[Editors' note: further revisions were requested prior to acceptance, as described below.]

Reviewer #2:[...] Comment #1: Regarding the protein sequence overlap and association with function, I suggest to add a short description of the Results section (1-2 sentences) and the figure in the supplementary material.

We have added a short paragraph (paragraph four in subsection "Functional annotations of alternative proteins") and Figure 11—figure supplement 3.

Reviewer #3:[...] As I mentioned in essential revision point 8 and 9, which the authors also agreed, co-conservation or domain sharing does not mean functional interaction/cooperation. The co-conservation or domain sharing evidence in this paper was very weak and did not support the conclusion, functional cooperation. In the other way, even the few cases mentioned in the paper which altORFs and annotated ORFs did functionally interact with each other do not show co-conservation or domain sharing. Simply changing "functional cooperation" to "functional relationship" makes no difference. The relevant parts in the manuscript should be removed, and the Title and Abstract should also be edited accordingly. Actually, the rest of the work is already very significant and solid.

As requested by the reviewer, we have completely removed the co-conservation analysis and corrected the Title and the Abstract accordingly. Since the presence of identical domains in alternative/reference protein coded by the same genes is part of the functional annotation, an important goal of the manuscript, we left this observation in the Results section; however, we removed the term "domain sharing".

We only briefly speculate about possible cooperation in the Discussion (Two sentences at the end of the first paragraph; one sentence in the last paragraph).